# Relative Geometry of Neural Forecasters: Linking Accuracy and Alignment in Learned Latent Geometry

**Deniz Kucukahmetler**                                               *kucukahm@cbs.mpg.de*
*Max Planck Institute for Human Cognitive and Brain Sciences, Leipzig, Germany*
*School of Embedded Composite Artificial Intelligence (SECAI), Dresden/Leipzig, Germany*

**Maximilian Jean Hemmann**                                          *maximilian@jeanm.de*
*Leipzig University, Leipzig, Germany*

*Julian Mosig von Aehrenfeld**                                   *julianvonmosig@gmail.com*
*Leipzig University, Leipzig, Germany*

*Maximilian Amthor**                                          *amthormaximilian@gmail.com*
*Leipzig University, Leipzig, Germany*

*Christian Deubel**                                             *christian.deubel@gmail.com*
*Leipzig University, Leipzig, Germany*

†**Nico Scherf**                                                       *nscherf@cbs.mpg.de*
*Max Planck Institute for Human Cognitive and Brain Sciences, Leipzig, Germany*
*Center for Scalable Data Analytics and Artificial Intelligence (ScaDS.AI), Dresden/Leipzig, Germany*

†**Diaaeldin Taha**                                                      *taha@mis.mpg.de*
*Max Planck Institute for Mathematics in the Sciences, Leipzig, Germany*

**Reviewed on OpenReview:** *https://openreview.net/forum?id=t4stf5Gafz*

## Abstract

Neural networks can accurately forecast complex dynamical systems, yet how they internally represent underlying latent geometry remains poorly understood. We study neural forecasters through the lens of representational alignment, introducing anchor-based, geometry-agnostic relative embeddings that remove rotational and scaling ambiguities in latent spaces. Applying this framework across seven canonical dynamical systems—ranging from periodic to chaotic—we reveal reproducible family-level structure: multilayer perceptrons align with other MLPs, recurrent networks with RNNs, while transformers and echo-state networks achieve strong forecasts despite weaker alignment. Alignment generally correlates with forecasting accuracy, yet high accuracy can coexist with low alignment. Relative geometry thus provides a simple, reproducible foundation for comparing how model families internalize and represent dynamical structure.‡

## 1 Introduction

Neural forecasters— recurrent neural networks (RNNs), transformers, and reservoirs are now routinely deployed to model complex, time-evolving phenomena across science and engineering. While forecasting performance is well studied, the geometry of learned propagated latent states—and how it varies across model

---

*Equal contribution.
†Equal supervision.
‡A shorter companion version of this work appears in the GTML (Geometry, Topology, and Machine Learning) 2025 workshop.

families—remains underexplored. As their use widens, it becomes essential to understand *how* these forecasters internally represent dynamical systems and whether those internal mechanisms align with human goals such as stability, interpretability, and transfer. A persistent obstacle is that latent spaces learned by different runs or model families are not directly comparable: coordinates can rotate, scale, shear, or even undergo more subtle geometric shifts with negligible effect on task loss but large effects on representational geometry. As a result, naive cross-model comparisons can be unstable and inconclusive (Figure 1 absolute latents).

Existing alignment tools only partially address this issue. Representational Similarity Analysis (RSA) (Kriegeskorte et al., 2008) captures pairwise relational structure but remains sensitive to the geometry of the distance matrix and sampling effects; Procrustes alignment assumes an approximately isometric map between spaces and often requires careful pairing; Centered kernel alignment (CKA) Kornblith et al. (2019) improves robustness to some transformations, but can still depend on dataset sampling, layer scaling, and kernel choices. Collectively, these limitations complicate systematic studies of how different model families encode dynamics and how those encodings relate to forecasting performance.

We reuse a geometry-agnostic alternative based on *relative embeddings* Moschella et al. (2023): anchor-based, extrinsic representations that index each point by its vector of similarities to a fixed set of anchors. By construction, these representations quotient out global rotations and scalings, are straightforward to compute, and yield a common coordinate system in which latent spaces from different seeds, layers, and model families can be compared directly. We apply this approach for an empirical analysis of encoder–propagator–decoder neural forecasters for dynamical systems.

This perspective matters for two reasons. First, it enables us to quantify representational families of neural forecasters—that is, which models converge to similar relational structures even when their raw latent geometries differ. Second, it links representation to utility: we find systematic patterns in how multilayer perceptrons, recurrent networks, transformers, and echo-state networks organize dynamical information, and show that our alignment signal carries practical information about forecasting accuracy. Notably, high predictive accuracy can coexist with low cross-forecaster alignment—especially in transformers—highlighting a gap between performance and representational agreement that standard metrics overlook. By aligning latent spaces through anchor-based relative embeddings, we expose reproducible family-level geometry across forecasters and offer a reproducible framework for studying how neural networks internalize dynamical structure.

We evaluate three neural model families—multilayer perceptrons (MLPs), recurrent neural networks (RNNs), and transformers (TF)—together with their Koopman- (K-) and Neural Ordinary Differential Equation (NODE, N-)–augmented variants, and an Echo State Network (ESN) as a no–backpropagation-through-time (no-BPTT) reference model. Code is available at `https://github.com/denizkucukahmetler/relative-geometry-neural-forecasting`.

**Contributions.**

- We reuse the relative-embedding alignment framework of Moschella et al. (2023) to neural forecasting of dynamical systems, yielding geometry-agnostic, anchor-based latent representations that are directly comparable across forecasters. Within this framework, we train forecasters end-to-end on relative representations and demonstrate cross-family latent stitching between MLP and transformer encoders and decoders.

- We conduct an extensive empirical study spanning seven canonical systems (continuous and discrete; periodic, quasi-periodic and chaotic) and three model families and a no-BPTT baseline.

- We uncover consistent family-level alignment patterns and characterize their relationship to forecasting error. We show that high predictive accuracy can coexist with low alignment—most prominently in transformers and ESNs—highlighting the limits of task loss alone and motivating representation-aware evaluation.

Together, these results suggest that anchor-based relative embeddings provide a simple, scalable basis for reproducible representation science in neural forecasting, enabling more faithful comparisons across seeds,

layers, and model families and offering new insights into how different model families internalize dynamical structure.

**Scope clarification.**   Throughout this work, alignment with the "true system" refers exclusively to alignment with the relative representation of observed trajectories under a shared anchor set, not to recovery of the system's governing equations, physical state variables, or dynamical invariants.

**Learning objective and interpretation.**   All models in this study are trained solely to minimize forecasting loss. Representational alignment is used as an *analysis tool*, not as an assumed or enforced consequence of the training objective. Observed alignment—or lack thereof—with the ground-truth relative representation reflects architectural inductive biases and task-induced representations, rather than evidence that minimizing forecasting loss recovers the underlying system dynamics. A central empirical finding of this work is precisely the divergence between forecasting accuracy and representational alignment, most prominently in transformers and ESNs.

## 2   Related Work

**Dynamical systems.**   Dynamical systems theory, from Poincaré's recurrence to modern hyperbolic dynamics, provides the mathematical backbone for modeling time-evolving processes, (Arnold, 1989; Katok & Hasselblatt, 1995; Strogatz, 2018). Compact, low-dimensional models such as the Lorenz-63 attractor (Lorenz, 1963) and the logistic map (May, 1976) famously revealed sensitive dependence on initial conditions and the geometry of strange attractors (Ruelle, 1978). Variants, including the higher-dimensional Lorenz-96 system (Lorenz, 1996), the Hamiltonian double pendulum, and the Hopf normal form, have since become canonical benchmarks for testing data-driven approaches. In fluid mechanics, proper orthogonal decomposition (POD) reductions of the cylinder wake (Brunton et al., 2016) serve as a tractable proxy for the Navier-Stokes equations. These systems are now present in most neural forecasting benchmarks: reservoir computers (Pathak et al., 2017; Matzner & Mráz, 2025), back-propagating RNNs (Vlachas et al., 2020), physics-informed latent ODEs (Raissi et al., 2019), and Koopman autoencoders (Lusch et al., 2018) are all routinely evaluated on one or two of them. We adopt the full suite – Lorenz-63, logistic map, Hopf oscillator, double pendulum, and POD-wake – thereby spanning periodic, quasi-periodic and chaotic regimes in both continuous and discrete time. This variety allows us to study how representation alignment behaves under qualitatively different underlying flows.

**Neural forecasting.**   Modeling and forecasting the evolution of dynamical systems is a cornerstone of scientific inquiry. Methods for dynamical system forecasting cover a wide spectrum, with first-principles modeling (Strogatz, 2018; Arnold, 1989) and data-driven modeling as two extremes. In the latter approach, which has gained popularity due to the availability of data and computing resources, the learned latent geometry of the system are learned directly from observations. Foundational work in nonlinear time-series analysis demonstrated this possibility by reconstructing system dynamics from data (Takens, 1981; Kantz & Schreiber, 2004). Today, this tradition is dominated by a diverse family of "neural forecasters," including RNNs (Hochreiter & Schmidhuber, 1997), transformers (Vaswani et al., 2017), Neural Ordinary Differential Equations (Chen et al., 2018), and forecasters inspired by Koopman operator theory (Lusch et al., 2018). Our study is situated within this data-driven context.

**Representational alignment.**   Foundational work in neuroscience on representational similarity analysis (RSA) provided a framework by comparing activity patterns by analyzing their distance matrices (Kriegeskorte et al., 2008). In machine learning, similar methods are used, like Procrustes analysis, which seeks an optimal rotational alignment between two sets of points (Gower, 1975; Schönemann, 1966), and, more recently, centered kernel alignment (CKA), which has become a standard for comparing neural representations across different initializations and model families (Kornblith et al., 2019; Ding et al., 2021). Other methods aim to create structured mappings between latent spaces using techniques like topological conjugation (Bizzi et al., 2025).

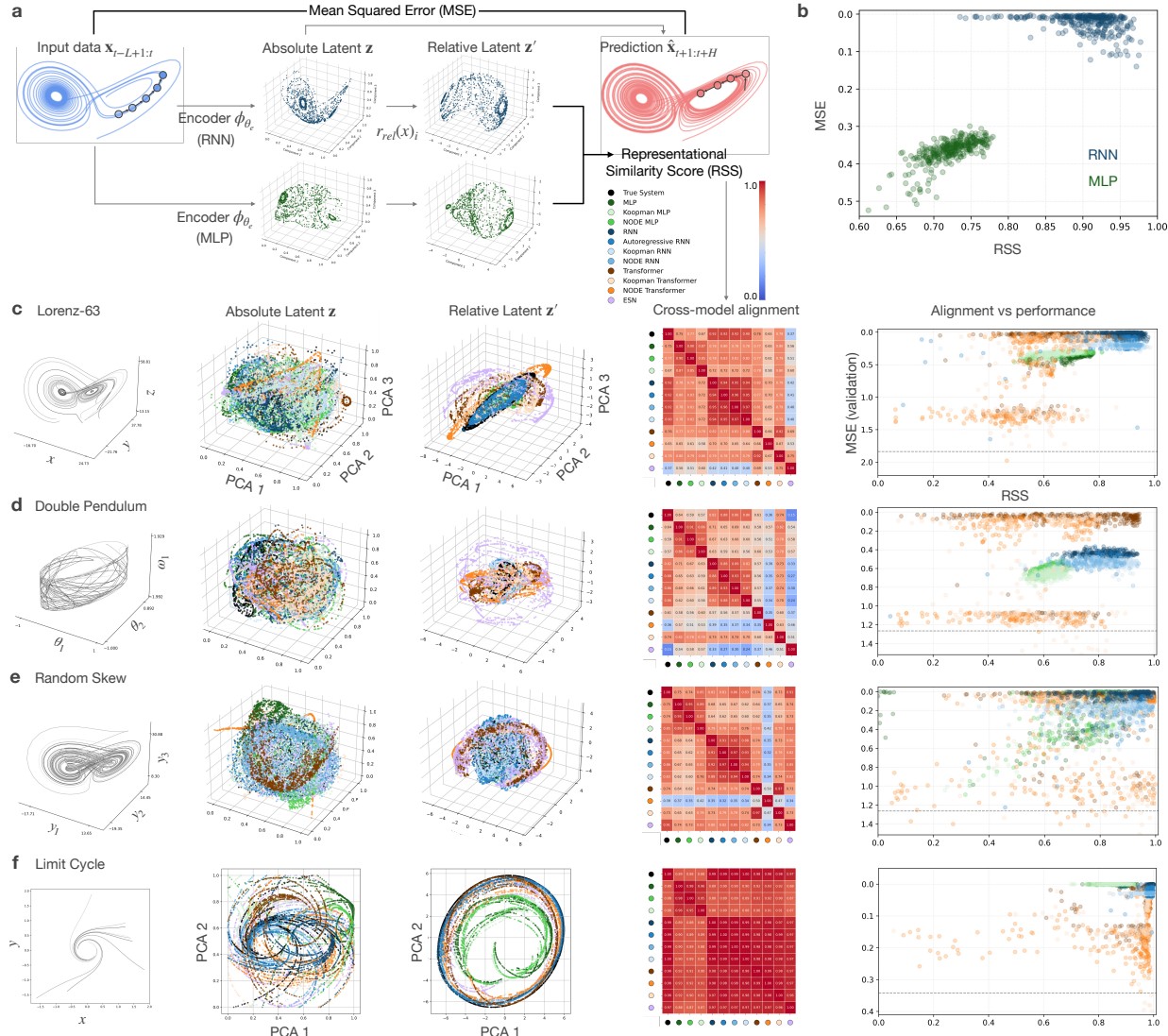

Figure 1: **Relative embeddings reveal consistent geometric structure across model families while removing rotational and scaling ambiguities.** (a) Encoder–propagator–decoder forecasters take an input window of $L$ past states $\mathbf{x}_{t-L+1:t}$, embed it into a latent vector $\mathbf{z}$, and decode a prediction of the next $H$ states $\widehat{\mathbf{x}}_{t+1:t+H}$. To compare different forecasters, we compute absolute latent embeddings from data, transform them into anchor-based relative embeddings using Moschella et al. (2023), and quantify alignment between forecasters using representational similarity scores. (b) Alignment–performance endpoints after training for RNNs (blue) and MLPs (green). RNNs achieve higher representational similarity and prediction accuracy (MSE), while MLPs show a clearer correlation between alignment and performance across seeds. (c-f) Example systems: Lorenz–63 (c), double pendulum (d), random skew (e), limit cycle (f). Columns display system trajectories, absolute embeddings (PCA; two or three principal components depending on dimensionality), relative embeddings (PCA), cross-forecaster similarity heatmaps averaged over five seeds—ordered as True System, MLP, Koopman MLP, NODE MLP, RNN, Autoregressive RNN, Koopman RNN, NODE RNN, Transformer, NODE Transformer, Koopman Transformer, and ESN; —and alignment–performance scatter plots across hyperparameter settings. Additional systems are shown in Appendix Figure 4 and 5.

**Relative representations.** In this work, we adopt a related but more direct approach that was first applied to computer vision models: anchor-based relative embeddings, which establish a standardized relational coordinate system to make latent spaces directly comparable (Moschella et al., 2023). Instead of defining a point's identity by its absolute coordinates, this technique represents it relationally—through its vector of similarities to a fixed set of anchor points—thus overcoming geometric ambiguities in latent spaces.

Building on this foundation, recent works have generalized relative representations. Anchor-based methods have been used to merge multiple latent spaces into a single aggregated one that preserves each space's geometry, akin to fusing several maps into a unified atlas (Crisostomi et al., 2023). This principle of latent-space stitching extends to other domains: unimodal vision models can be stitched into a multimodal model without additional training (Norelli et al., 2023), while RL agent policies can be stitched to form new agents for unseen visual–task combinations (Ricciardi et al., 2024). Further refinements add topological and geometric stability for zero-shot stitching (García-Castellanos et al., 2024), and even show that simple linear transformations can rival anchor-based methods in latent space alignment (Lähner & Moeller, 2024).

Building on these insights, Latent Functional Maps (Fumero et al., 2025) introduce a spectral formulation that enables robust cross-space transfer. Similarly, Maiorca et al. (2023) estimate direct transformations between latent spaces without training decoders on relative representations. Lastly, Cannistraci et al. (2024) propose constructing product latent spaces composed of multiple invariant components, each induced by distinct similarity functions.

Anchor-based relative representations are closely related to landmark-based methods–long used in dimensionality reduction, clustering, and kernel learning Faloutsos & Lin (1995) De Silva & Tenenbaum (2004) Oglic & Gärtner (2017) Chen & Cai (2011) Liu et al. (2010). Using landmarks, a point is represented as the distance or similarity to a fixed set of landmarks. Anchor-based approaches extend this to neural latent spaces.

This study considers such relational and anchor-based techniques within the domain of dynamical systems forecasting, where comparable latent spaces are essential for analyzing, aligning, and transferring representations across contexts.

## 3 Method

### 3.1 Representational alignment experiment design

**The representational alignment framework.** Following Sucholutsky et al. (2023), a *representational alignment experiment* consists of *data*, *systems* (models, in our case), *measurements*, *embeddings* and a *similarity metric*. We spell out these ingredients for our study:

- **Data**: simulated trajectories from seven dynamical systems (Sections 3.2 and 4.1)

- **Neural forecasters**: each trained encoder–propagator–decoder model instance (seed, model) (Sections 3.3 and 4.2)

- **Measurement operator** $m$: the encoder's latent vector $\mathbf{z} = \phi_{\theta_e}(\mathbf{x}_{t-L+1:t})$ (Section 3.4)

- **Embeddings**: anchor-based relative embeddings $r(x)$ obtained by z-scored distances (Section 3.5)

- **Similarity metric**: cosine, rank and T1 similarity of two relative embeddings (Section 3.6)

We provide a summary of the notation used in this section in Table 1.

**Representational alignment task.** In the sense of Sucholutsky et al. (2023), this work primarily addresses the *measuring representational alignment* task: we quantify pairwise similarity across encoder initializations and model families and examine how that similarity relates to forecasting loss. We also explore aspects of *bridging* through cross-family latent stitching, though alignment remains substantially stronger within than across families. Developing effective bridging mappings and alignment-driven training interventions is left to future work.

## 3.2 Data: Trajectories of dynamical systems

A *dynamical system* is a triple $(T, X, \Phi)$ in which $T$ is an *additive monoid* that plays the role of time (e.g., $T = \mathbb{R}$ for continuous time, or $T = \mathbb{Z}$ for discrete time), $X$ is a non-empty *state space*, and $\Phi : T \times X \to X$ is the *evolution map* (also called the *flow*) satisfying $\Phi(0, x) = x$ and $\Phi(t_2, \Phi(t_1, x)) = \Phi(t_1 + t_2, x)$ for all admissible $t_1, t_2 \in T$ and $x \in X$. For a fixed initial state $x \in X$, the curve $\Phi_x : T \to X$, $t \mapsto \Phi(t, x)$ is the *trajectory* (or *orbit*) through $x$; its image $\gamma_x = \{\Phi(t, x) \mid t \in T\}$ is the set of states visited over time.

## 3.3 Model: Neural forecasters

Given a window of $L$ past states $\mathbf{x}_{t-L+1:t} \in \mathbb{R}^{L \times d}$, the aim of the *forecasting task* is to predict the next $H$ steps $\mathbf{x}_{t+1:t+H} \in \mathbb{R}^{H \times d}$. In this study, we employ encoder–propagator–decoder neural networks $g = \psi_{\theta_d} \circ \mathcal{P}_\Theta \circ \phi_{\theta_e}$ as forecasters: the encoder $\phi_{\theta_e} : \mathbb{R}^{L \times d} \to \mathbb{R}^k$ maps the input slice to a latent vector, the propagator $\mathcal{P}_\Theta : \mathbb{R}^k \to \mathbb{R}^k$ evolves that latent, and the decoder $\psi_{\theta_d} : \mathbb{R}^k \to \mathbb{R}^{H \times d}$ produces the $H$-step prediction. Parameters are trained to minimise a forecasting loss $\mathcal{L}_{\text{pred}}$ (we use mean-squared error (MSE)) over trajectories drawn from the unknown dynamical system. We use the term *forecaster* for a trained model instance (architecture, hyperparameters, and learned weights), and *model* for the corresponding untrained architecture or configuration.

## 3.4 Measurements: Latent representations

The encoder $\phi_{\theta_e} : \mathbb{R}^{L \times d} \to \mathbb{R}^k$ maps an input segment to a latent vector $\mathbf{z} \in \mathbb{R}^k$. Training the same model with different random seeds, or swapping to a different model, yields a family of encoders $\{\phi_{\theta_e^{(s)}}^{(s)}\}_{s=1}^S$ whose latent space alignment is the subject of this study.

## 3.5 Embeddings: Anchor-based relative embeddings

Let $\mathcal{V} = \{\mathbf{z}_j\}_{j=1}^N \subset \mathbb{R}^k$ denote the set of latent representations obtained by applying the encoder to input windows:

$$\mathbf{z}_j = \phi_{\theta_e}(\mathbf{x}_{t_j - L + 1 : t_j}), \qquad \mathbf{x}_{t_j - L + 1 : t_j} \in \mathbb{R}^{L \times d}.$$

We select a subset $\mathcal{A} = \{\mathbf{a}_i\}_{i=1}^m \subset \mathcal{V}$ as *anchors*. Let $\text{sim} : \mathbb{R}^k \times \mathbb{R}^k \to \mathbb{R}$ be a similarity function.

**Relative embeddings via z-scoring.** Each encoder produces latent vectors $\mathbf{z}_j = \phi_{\theta_e}(\mathbf{x}_{t_j - L + 1 : t_j}) \in \mathbb{R}^k$, which are first z-scored feature-wise across the dataset. A fixed subset $\mathcal{A} = \{\mathbf{a}_i\}_{i=1}^m \subset \{\mathbf{z}_j\}_{j=1}^N$ serves as anchors, and each normalized latent yields a *relative embedding*

$$\mathbf{z}' = \mathbf{r}_{\text{rel}}(\mathbf{z}) = \big(\text{sim}(\mathbf{z}, \mathbf{a}_1), \ldots, \text{sim}(\mathbf{z}, \mathbf{a}_m)\big).$$

where $\text{sim}(\cdot, \cdot)$ denotes a similarity function introduced in Section 3.6. This produces, for each forecaster, a matrix $\mathbf{R}_{\text{rel}} \in \mathbb{R}^{N \times m}$ whose rows correspond to data points and columns to anchors.

## 3.6 Similarity metric: Similarity of two encoders

We quantify the similarity between two encoders $\phi_{\theta_e^{(1)}}^{(1)}$ and $\phi_{\theta_e^{(2)}}^{(2)}$ over a dataset $\mathcal{V}$ using three complementary metrics: *cosine similarity*, *rank similarity*, and *T1 score*. Each of these captures different aspects of agreement between the encoders' relative embeddings $\mathbf{z}'^{(1)}$ and $\mathbf{z}'^{(2)}$.

**Cosine similarity.** The representational similarity score (RSS) is defined as the mean cosine similarity of the relative embeddings:

$$\alpha_{\cos}\left(\phi_{\theta_e^{(1)}}^{(1)}, \phi_{\theta_e^{(2)}}^{(2)}; \mathcal{V}\right) = \frac{1}{|\mathcal{V}|} \sum_{\mathbf{z} \in \mathcal{V}} \frac{\langle \mathbf{z}'^{(1)}, \mathbf{z}'^{(2)} \rangle}{\|\mathbf{z}'^{(1)}\|_2 \, \|\mathbf{z}'^{(2)}\|_2},$$

where $\mathbf{z}'^{(s)} = \mathbf{r}_{\text{rel}}^{(s)}(\mathbf{z})$ denotes the relative embedding induced by encoder $s \in \{1, 2\}$, and $\alpha_{\cos}$ denotes the RSS used throughout our experiments. **Rank similarity** and **T1 score** are defined in Appendix B.

## 3.7 Stitching

We define a stitched model as the composition of an *encoder* that produces a latent, a fixed *relative* transformation with a global anchor set $\mathcal{A}$ (Moschella et al., 2023; Crisostomi et al., 2023), and a task-specific *propagator–decoder* that operates in the $|\mathcal{A}|$-dimensional relative space. Concretely, the encoder $\phi_{\theta_e} : \mathbb{R}^{L \times d} \to \mathbb{R}^k$ maps an input window to a latent vector, which is mapped to a relative representation via cosine similarities to $\mathcal{A}$ (z-scored per anchor) (Moschella et al., 2023). The propagator $\mathcal{P}_\Theta : \mathbb{R}^{|\mathcal{A}|} \to \mathbb{R}^{|\mathcal{A}|}$ and decoder $\psi_{\theta_d} : \mathbb{R}^{|\mathcal{A}|} \to \mathbb{R}^{H \times d}$ are trained end-to-end in this relative space.

Crucially, because all decoders consume the same relative representation, any trained decoder can be *stitched* to any trained encoder without additional training (Moschella et al., 2023; Norelli et al., 2023; Ricciardi et al., 2024). We evaluate stitching by swapping encoders and decoders across families and reporting $H$-step MSE. For comparison, we also train *absolute* variants that omit the relative transform; such models can only be stitched when latent dimensions match and are generally less stable (Lähner & Moeller, 2024; Maiorca et al., 2023). Recurrent model families (e.g., RNN/ESN) are excluded from cross-family stitching due to their dependence on hidden state, which is not provided by non-recurrent encoders.

# 4 Experimental setup

## 4.1 Dynamical systems

**Dynamical systems considered.** We evaluate neural forecasters on a collection of canonical dynamical systems spanning discrete and continuous time, dissipative and conservative dynamics, and low- to moderately high-dimensional state spaces (details in Appendix D). For clarity, we briefly summarize the qualitative dynamical regime represented by each system.

- **Lorenz–63 (chaotic, dissipative).** A three-dimensional continuous-time system with a strange attractor, characterized by sensitive dependence on initial conditions and strong nonlinear coupling. It represents a classical example of low-dimensional dissipative chaos.

- **Stable limit cycle system (periodic).** A two-dimensional radial–spiral flow whose trajectories converge to a closed orbit. This system provides a simple nonlinear periodic regime with smooth and predictable long-term behavior.

- **Double pendulum (Hamiltonian chaos).**[†] A four-dimensional energy-conserving mechanical system exhibiting chaotic motion due to nonlinear interactions. Unlike dissipative chaotic systems, trajectories evolve on a conserved-energy manifold.

- **Hopf normal form (nonlinear periodic).** A two-dimensional system undergoing a supercritical Hopf bifurcation, producing a single-frequency stable limit cycle. It represents weakly nonlinear periodic dynamics near the onset of oscillations.

- **Logistic map (discrete chaos).** A one-dimensional discrete-time system at a parameter value yielding chaotic behavior. Its stretching-and-folding dynamics provide a canonical example of discrete-time chaos distinct from continuous flows.

- **POD wake (reduced spatiotemporal dynamics).** A three-mode Proper Orthogonal Decomposition of a fluid wake, capturing coherent structures of an underlying high-dimensional spatiotemporal flow. The resulting reduced-order system exhibits multi-scale temporal variability inherited from turbulent dynamics.

- **Skew-product system (high-dimensional coupled chaos).** A six-dimensional system formed by weakly coupling multiple chaotic subsystems Lai et al. (2025). This construction introduces interacting but partially separable chaotic modes, increasing effective dynamical complexity while retaining interpretable structure.

---

[†]quasi-periodic for low amplitude

- **iEEG recordings (real neural dynamics; external test case).** Intracranial EEG (iEEG) time series from a human subject Ghosh (2024), providing a high-dimensional, noisy, and partially observed real-world dynamical system. We use this dataset as an external validation to assess whether the relative geometric trends observed on synthetic systems transfer to empirical neural data, focusing on representational geometry rather than neuroscientific interpretation (details in Appendix F).

Unless noted otherwise, all *synthetic* dynamical systems provide trajectories from 30 distinct initial conditions, each of length $T$=500 time steps. These trajectories are equally split into training, validation, and test sets, and a sliding window is used for data augmentation. All channels are z-scored using statistics computed on the training split, and no external noise is added. Data generation scripts for the synthetic systems are provided in the `GitHub repository`.

## 4.2 Neural forecasters

Given an input window of $L$ past states $\mathbf{x}_{t-L+1:t} \in \mathbb{R}^{L \times d}$, the forecasting task is to predict the next $H$ steps $\mathbf{x}_{t+1:t+H} \in \mathbb{R}^{H \times d}$. All encoder–decoder models share the factorization

$$\widehat{\mathbf{x}}_{t+1:t+H} = \psi_{\theta_d}\big(\mathcal{P}_\Theta\big(\phi_{\theta_e}(\mathbf{x}_{t-L+1:t})\big)\big),$$

with encoder $\phi_{\theta_e} : \mathbb{R}^{L \times d} \to \mathbb{R}^k$, optional latent propagator $\mathcal{P}_\Theta : \mathbb{R}^k \to \mathbb{R}^k$, and decoder $\psi_{\theta_d} : \mathbb{R}^k \to \mathbb{R}^{H \times d}$. We instantiate this framework with MLP, RNN, and transformer families, together with their K-, N-, and A- variants defined in Section 1, and include an ESN baseline described at the end of this section.

**Latent state propagation.** To impose temporal structure in the latent space, the encoder maps the input window to an initial latent state $\mathbf{z}_0 = \phi_{\theta_e}(\mathbf{x}_{t-L+1:t})$, which is then evolved forward for $H$ steps through a latent propagator $\mathcal{P}_\Theta$. The terminal latent state $\mathbf{z}_H$ is decoded to produce the forecast, $\widehat{\mathbf{x}}_{t+1:t+H} = \psi_{\theta_d}(\mathbf{z}_H)$. We consider the following choices for $\mathcal{P}_\Theta$:

$$\textbf{Identity:} \quad \mathbf{z}_H = \mathbf{z}_0,$$

$$\textbf{Neural-ODE:} \quad \dot{\mathbf{z}} = f_\Theta(\mathbf{z}, t), \quad \mathbf{z}_H = \mathrm{RK45}\big(f_\Theta, \mathbf{z}_0, H\Delta t\big),$$

$$\textbf{Koopman (linear):} \quad \mathbf{z}_{k+1} = \boldsymbol{K}\,\mathbf{z}_k, \quad k = 0, \dots, H-1, \boldsymbol{K} \in \mathbb{R}^{k \times k}.$$

In the identity case, the model reduces to a standard one-shot encoder–decoder forecaster, $\widehat{\mathbf{x}}_{t+1:t+H} = \psi_{\theta_d}\big(\phi_{\theta_e}(\mathbf{x}_{t-L+1:t})\big)$. A summary of the propagators is provided in Table 2.

**Transformer forecaster.** Our transformer forecaster follows a standard encoder–decoder architecture with multi-head self-attention, feed-forward layers, residual connections, and sinusoidal positional encodings (Vaswani et al., 2017). In contrast to recurrent models, the transformer does *not* maintain or propagate an explicit latent state across forecast steps. Instead, it performs *block (one-shot) multi-step prediction*: the encoder summarizes the input window into a latent representation, and the decoder predicts the entire $H$-step forecast in a single forward pass. Causal masking is applied in the decoder to preserve temporal ordering within the prediction horizon, but this masking does not induce a recurrent hidden-state evolution. As a result, the transformer's internal representations need not form smooth latent trajectories over forecast time, which distinguishes it from RNN- and propagator-based forecasters.

**Reservoir baseline.** The echo-state network does not use an encoder–decoder split. A fixed sparse reservoir updates via $\mathbf{r}_{k+1} = \tanh(\boldsymbol{W}\mathbf{r}_k + \boldsymbol{U}\mathbf{x}_k)$; all $L$ inputs are retained (no wash-out). Only the linear read-out $\boldsymbol{W}_{out}$ is fitted by ridge regression, providing a no-BPTT reference.

Table 1: Summary of notation used for trajectories, latent states, anchors, and representational similarity.

| Symbol | Description |
|---|---|
| *Trajectories and dynamical systems (Section 3.2)* | |
| $T$ | Time index set (discrete or continuous). |
| $X$ | State space of the dynamical system. |
| $\Phi : T \times X \to X$ | Evolution map of the dynamical system. |
| $x \in X$ | Initial condition or system state. |
| $\Phi_x(t)$ | Trajectory starting from $x$. |
| $\gamma_x$ | Image of the trajectory $\Phi_x$. |
| *Latent states and forecasting model (Section 3.3, 3.4)* | |
| $L$ | Input (context) window length. |
| $H$ | Forecast horizon. |
| $x_{t-L+1:t}$ | Input window of observed states. |
| $z$ | Latent state produced by the encoder. |
| $\phi_{\theta_e}$ | Encoder mapping inputs to latent space. |
| $P_\Theta$ | Latent propagator (identity, NODE, or Koopman). |
| $z_H$ | Latent state after $H$ propagation steps. |
| $\psi_{\theta_d}$ | Decoder mapping latent states to predictions. |
| *Anchors and relative latent representations (Section 3.5, Section 3.6)* | |
| $N$ | Number of encoded input windows used to construct the latent set $V$. |
| $V = \{z_j\}_{j=1}^N$ | Set of latent states obtained by encoding $N$ input windows for model $m$. |
| $A = \{a_i\}_{i=1}^m$ | Anchor set, a subset of $V$. |
| | (per model; anchors correspond to the same sampled input windows across models) |
| $a_i$ | $i$-th anchor latent vector. |
| $m$ | Number of anchors. |
| $z'$ | Relative embedding of latent $z$ with respect to anchors. |
| $z'_i$ | $i$-th coordinate of $z'$. |
| $\text{sim}(\cdot, \cdot)$ | Similarity function between latent vectors. |
| $\alpha_{\cos}$ | Mean cosine similarity between two models relative embeddings |

Table 2: Encoder–Propagator–Decoder decomposition across model families.

| Model | Encoder | Propagator | Decoder |
|---|---|---|---|
| MLP | MLP (feed-forward) | Identity ($\mathcal{P}(\mathbf{z}) = \mathbf{z}$) | MLP (feed-forward) |
| RNN | RNN (GRU) | Identity | RNN (GRU) |
| A-RNN | RNN (GRU, autoregressive) | Identity | RNN (GRU, autoregressive) |
| Transformer (TF) | Transformer (causal attention) | Identity | Transformer (causal attention) |
| N–MLP, RNN, TF | Same as base model | NODE: $\dot{\mathbf{z}} = f_\Theta(\mathbf{z}, t)$ | Same as base model |
| K–MLP, RNN, TF | Same as base model | Linear: $\mathbf{z}_{k+1} = \boldsymbol{K}\mathbf{z}_k$ | Same as base model |
| ESN | None (random reservoir) | $\mathbf{r}_{k+1} = \tanh(\boldsymbol{W}\mathbf{r}_k + \boldsymbol{U}\mathbf{x}_k)$ | Linear readout |

## 4.3 Training details

**Optimisation.** Adam optimiser, step size $10^{-3}$, exponential decay factor 0.95. Early stopping (patience 20) monitors validation MSE. The hyperparameter tuning settings and results are reported at `compiled_results.csv`, and the tuned parameters used in the reported experiments (i.e., cross-forecaster alignment, benchmarking, and perturbation experiments) are provided at `best_model_parameters.csv` for all trainable models, and at `esn_hyperparams.csv` for the ESN.

**Key hyperparameters.** For each dynamical system and each model variant (excluding ESNs), we selected the best configuration from the hyperparameter search based on validation performance. Across all MLP, Koopman-MLP, and NODE-MLP models, selected learning rates lay in the range $5 \times 10^{-4}$–$10^{-3}$. Batch size was typically 64, with NODE-MLP models consistently using batch size 32 across systems. Latent dimensionality ranged from 64 to 256, while hidden layer widths varied between 128 and 1024, depending on system complexity.

RNN, Koopman-RNN, and NODE-RNN models used encoder widths between 64 and 512 (most frequently 256), with 2–5 layers in both encoder and decoder. Latent dimensions for these models ranged from 32 to 128 across systems.

Transformer, Koopman-Transformer, and NODE-Transformer models consistently selected model dimensions between 128 and 384, using 2–8 attention heads. Batch size was 64 for Transformer and Koopman-Transformer models and 128 for NODE-Transformer models, with learning rates of either $10^{-3}$ or $5 \times 10^{-4}$ depending on the specific system–model combination. Dropout was applied selectively and most often set to 0.1.

No single architectural variant (identity, Koopman, or NODE) dominated uniformly across all systems; instead, different variants were preferred for different system–model combinations. Full hyperparameter ranges and corresponding validation performances are reported in `compiled_results.csv` and the best combinations producing the lowest validation MSE are reported in `best_model_parameters.csv`.

### 4.4 Evaluation metrics

Test performance (five random seeds) is reported using (i) mean-squared error (MSE), (ii) root-mean-squared error (RMSE), and (iii) mean absolute error (MAE). Each metric is computed per step and then averaged over the 50-step forecast.

## 5 Experimental Results

Table 3: Performance and Alignment for `lorenz`

| Model | Performance (↓) | | | Similarity (↑) | | |
|---|---|---|---|---|---|---|
| | MSE | MAE | RMSE | Cosine | Top-1 | Spearman $\rho$ |
| MLP | $0.3828 \pm 0.0080$ | $0.2804 \pm 0.0050$ | $0.6187 \pm 0.0064$ | $0.7148 \pm 0.0193$ | $0.3786 \pm 0.0025$ | $0.7121 \pm 0.0187$ |
| Koopman MLP | $0.3597 \pm 0.0288$ | $0.2824 \pm 0.0179$ | $0.5994 \pm 0.0245$ | $0.6372 \pm 0.0252$ | $0.3722 \pm 0.0048$ | $0.6344 \pm 0.0271$ |
| NODE MLP | $0.3684 \pm 0.0081$ | $0.2939 \pm 0.0176$ | $0.6070 \pm 0.0067$ | $0.7489 \pm 0.0088$ | $0.3928 \pm 0.0008$ | $0.7476 \pm 0.0115$ |
| RNN | $0.0096 \pm 0.0018$ | $0.0570 \pm 0.0073$ | $0.0979 \pm 0.0092$ | $\mathbf{0.9125 \pm 0.0128}$ | $0.5100 \pm 0.0162$ | $\mathbf{0.9031 \pm 0.0117}$ |
| Autoregressive RNN | $0.0422 \pm 0.0082$ | $0.1172 \pm 0.0106$ | $0.2045 \pm 0.0203$ | $0.9084 \pm 0.0124$ | $0.5084 \pm 0.0124$ | $0.8990 \pm 0.0150$ |
| Koopman RNN | $0.1259 \pm 0.0364$ | $0.2413 \pm 0.0262$ | $0.3520 \pm 0.0503$ | $0.8899 \pm 0.0167$ | $0.4952 \pm 0.0113$ | $0.8824 \pm 0.0167$ |
| NODE RNN | $0.1576 \pm 0.0486$ | $0.2629 \pm 0.0340$ | $0.3931 \pm 0.0624$ | $0.9053 \pm 0.0132$ | $\mathbf{0.5150 \pm 0.0113}$ | $0.8950 \pm 0.0142$ |
| Transformer | $\mathbf{0.0049 \pm 0.0014}$ | $0.0547 \pm 0.0085$ | $\mathbf{0.0693 \pm 0.0102}$ | $0.7263 \pm 0.0473$ | $0.4316 \pm 0.0167$ | $0.7295 \pm 0.0417$ |
| Koopman Transformer | $0.0129 \pm 0.0019$ | $0.0882 \pm 0.0056$ | $0.1132 \pm 0.0085$ | $0.7447 \pm 0.0365$ | $0.4208 \pm 0.0082$ | $0.7467 \pm 0.0337$ |
| NODE Transformer | $0.0244 \pm 0.0050$ | $0.1206 \pm 0.0111$ | $0.1556 \pm 0.0153$ | $0.6373 \pm 0.1201$ | $0.3398 \pm 0.0322$ | $0.6192 \pm 0.0800$ |
| ESN | $0.0102 \pm 0.0050$ | $\mathbf{0.0256 \pm 0.0046}$ | $0.0988 \pm 0.0241$ | $0.3435 \pm 0.0025$ | $0.3124 \pm 0.0022$ | $0.3388 \pm 0.0026$ |

We next evaluate how relative embeddings capture representational geometry across model families, systems, and training conditions, focusing on (i) cross-family alignment, (ii) its relationship to forecasting accuracy, and (iii) its stability under perturbations.

**Relative embeddings establish a shared representational space across model families.** Figure 1 illustrates that anchor-based *relative* embeddings reduce geometric arbitrariness (rotations, scalings) in latent spaces, making cross-forecaster comparisons more interpretable. With colors indicating distinct forecaster labels, the relative space reveals similarities and differences across forecasters in a common coordinate system. For completeness, we also quantitatively assessed cross-forecaster alignment in the original latent spaces, confirming substantial misalignment (Appendix Figure 11).

**Models form reproducible family-level alignment patterns.** Cross-forecaster similarity in Figure 1 (pairwise alignment heatmaps; cosine similarity of relative embeddings) reveals consistent family structure across systems: (i) in all systems, the *MLP family* (MLP, Koopman–MLP, NODE–MLP) forms a cluster; (ii) the *RNN family* (RNN, autoregressive RNN, Koopman–RNN, NODE–RNN) is well-aligned in all systems

*except* the Logistic Map (Appendix Figure 5), where alignment weakens; (iii) the ESN baseline exhibits noticeably lower alignment in Lorenz-63, double pendulum, and the random skew product; (iv) the *transformer family* tends to align less with other families— prominently in double pendulum, Lorenz-63 and random skew—suggesting a different inductive bias in how context is summarized for forecasting. Overall, these patterns indicate that architectural choices induce reproducible representational geometries within families, while some dynamics (e.g., Logistic Map) challenge specific families (RNNs). As an external validation, we additionally report preliminary results on a high-dimensional real-world iEEG forecasting task in Appendix F, where we observe qualitatively similar family-level alignment patterns across architectures.

**Forecast accuracy and representational alignment diverge across model families.**

To assess the alignment with the true system for each forecaster family more systematically, we trained multiple forecasters per dynamical system (Figure 1, Alignment vs. Performance column; single forecaster results in Appendix Figure 10) and plotted the final MSE against the alignment with the true system. For MLPs, performance is more strongly related with alignment. Transformers, on the other hand, exhibit higher variability: they achieve both the best and worst scores across seeds, and strong performance does not always coincide with strong alignment. However, they rarely appear in the bottom-left quartile of the plot, indicating that transformers typically do not show low performance and low alignment jointly. RNNs mostly cluster in the top-right quadrant, suggesting that they consistently attain both high performance and high alignment. Overall, across model families, we observe a general positive relationship between representational similarity and forecasting accuracy, although its strength varies by forecaster family and dynamical system (Figure 1).

Next we studied performance to alignment with the true system during *training*. We observe family-specific training trajectories (First column in Figure 2). *RNNs* begin with comparatively high alignment and remain stable through training across systems (except for Logistic Maps), while their test error decreases steadily. *MLPs* either show a similar pattern to RNNs (high alignment from the beginning, seen in all systems except Lorenz-63 and double pendulum) or start with lower alignment that increases as training proceeds (seen in Lorenz-63 and double pendulum), tracking improvements in error. *Transformers* display lower and more variable alignment across seeds, yet often achieve competitive or superior performance—frequently surpassing the MLP family and often rivaling RNN variants. This underscores that high alignment is *helpful but not strictly necessary* for strong forecasting: transformers can achieve good accuracy with a representational geometry that aligns less to the ground-truth relative space.

**Noise and input length differently affect representational stability across forecasters.** To evaluate the effects of practically relevant parameters such as input noise or available (input) sequence length ($L$), we measured both predictive performance and representational alignment with the ground-truth dynamics under varying noise levels and sequence lengths. For these experiments, we selected one representative forecaster from each major family and report results for MLP, transformer, and autoregressive RNN (Figure 2; see Appendix 6 for the remaining systems).

Increasing the noise level consistently degraded both alignment and forecasting accuracy, but impacts the forecasters differently. Across all dynamical systems, RNNs tended to lose representational similarity more rapidly than predictive performance with a nearly linear trend. In contrast, transformers show a more nonlinear behavior, where performance decreases steeply with noise in the Lorenz-63 and double pendulum systems. A similar pattern can be seen for MLPs for the limit cycle. The random skew system shows a qualitatively similar picture but the patterns are overall less clear in this case.

The effect of input length ($L$) varied across both forecasters and dynamical systems. In many cases, neither performance nor alignment changed dramatically with $L$ (like for transformers in Lorenz, random skew or limit cycle), though notable exceptions exist. RNN and transformers show a similar pattern: performance remained largely stable across systems, but alignment exhibited high variability for some systems (double pendulum and logistic-map for RNNs and double pendulum and POD-wake for transformers). By comparison, MLPs were more sensitive to longer input windows—their performance and alignment degraded with increasing $L$ in the Lorenz, double pendulum, limit cycle, POD-wake, and Hopf systems, while remaining stable in the random skew and logistic-map settings.

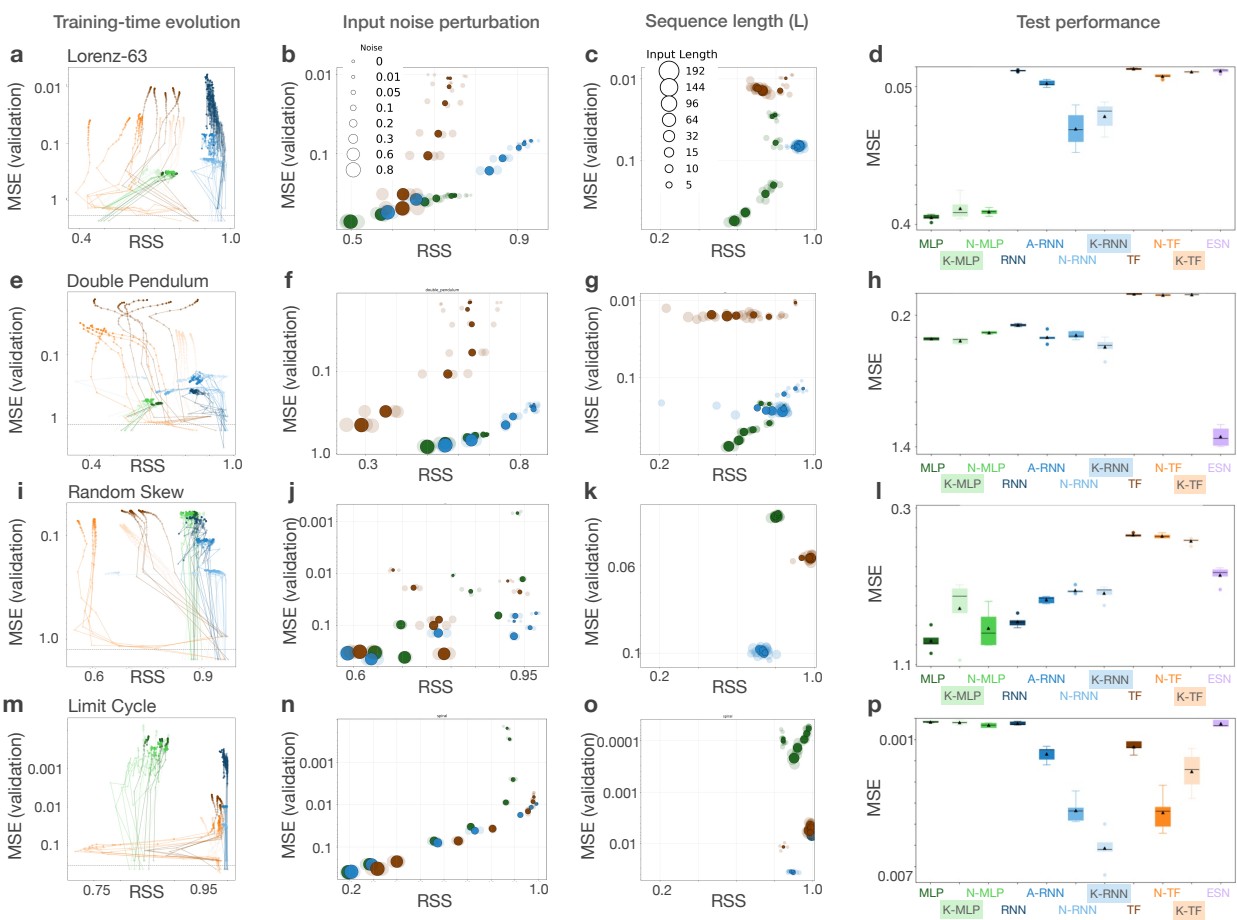

Figure 2: **Performance–alignment trade-offs across training, noise, and input conditions.** Columns show (a-m) training time evolution, (b-n) effects of input noise, (c-o) effects of sequence length *L*, and (d-p) test performance across model families (MLP, K-MLP: Koopman MLP, N-MLP: NODE MLP, RNN, A-RNN: Autoregressive RNN, K-RNN: Koopman RNN, N-RNN: NODE RNN, TF: Transformer, N-TF: NODE Transformer, K-TF: Koopman Transformer, ESN). Each point represents the mean squared error (MSE) and the representational similarity score (RSS) of a given a forecaster trained with a different random seed (color-coded by forecaster family; same-colored lines/points denote different initializations of the same forecaster). MLPs and RNNs exhibit consistent performance–alignment relationships, while transformers show larger variability; ESNs are excluded due to their no–backpropagation-through-time (no-BPTT) training. Increasing input noise consistently degrades both alignment and accuracy, whereas varying *L* produces system-dependent effects, highlighting differences in robustness across model families. Test results (d-p) indicate that no single family dominates across all dynamical systems. Results for additional systems in Appendix 6.

These results highlight that researchers interested in *geometric or representational stability*, rather than accuracy alone, should consider noise levels and input-length choices when selecting forecasting model families.

**Alignment estimates stabilize with increasing number of anchors.** We empirically assess how the number of anchors affects relative-representation alignment between our pretrained MLP–MLP forecaster and the ground-truth Lorenz-63 system by correlating their anchor–sample similarity matrices using the method Moschella et al. (2023). We swept the number of anchors $K \in \{1, 2, 3, 4, 5, 6, 8, 16, 32, 64, 128, 512, 800, 999\}$ and, for each $K$, repeated the estimation 30 times with independently sampled anchors (without replacement). The alignment estimates display an approximately constant mean once $K \geq 16$ (around $r \approx 0.74$), while the across-repeat variability decreases markedly with increasing $K$ (Appendix Figure 9). This vari-

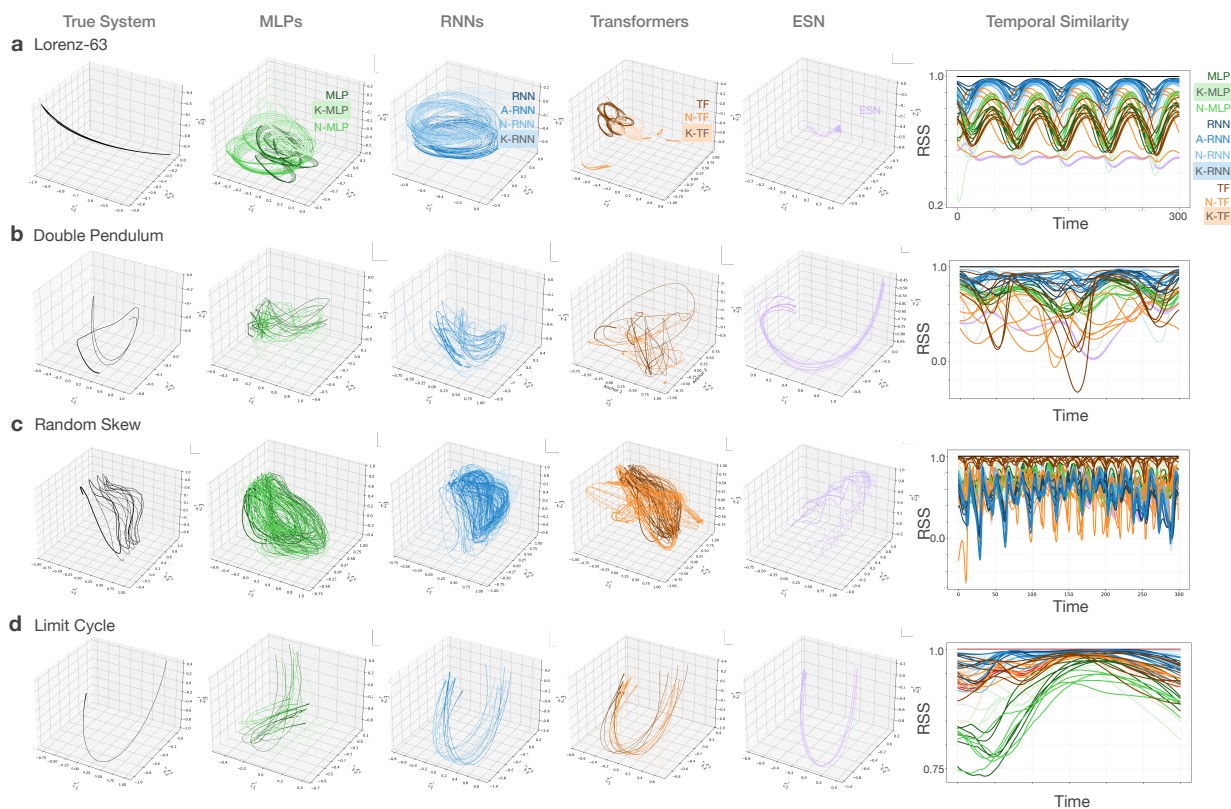

Figure 3: **Temporal evolution of representational alignment across dynamical systems.** Each row shows the true system (left), reconstructed trajectories from different model families (MLPs, RNNs, transformers, ESN; same colour coding as in Figure 2), and their temporal similarity profiles (right; line thickness encodes time, $T = 300$). For visualisation purposes we use relative coordinates $z_i'$ with respect to three anchor points (axes $z_1'$, $z_2'$, $z_3'$). For the Lorenz–63, double pendulum, and random skew systems, MLPs and RNNs maintain representations closely aligned with the true dynamics, whereas transformers and ESNs diverge. For the limit cycle and other periodic systems (see Appendix 7), all families capture similarly structured representations. The Logistic map is omitted due to its one-dimensional, contractive behavior.

ance reduction indicates convergence of the estimator as more anchors are used. Balancing stability and computation, we fixed $K = 80$ for all experiments except in the stitching experiment, where $K = 32$ was used.

As a random baseline (orange line in Appendix Figure 9), we drew distinct anchor sets for the forecaster and the true system. Under this mismatch, alignment was near zero across all $K$ ($r \approx 0$), confirming that the observed nonzero alignment with shared anchors reflects genuine representational correspondence rather than sampling artifacts.

As three anchors already yield a relatively reliable similarity estimate, this suggests that the relative coordinates can directly be used for (a randomized) low-dimensional visualization of embedded latent geometry.

**Alignment varies along trajectories and across model families.** So far we have computed representational alignment over all latent points. To assess how alignment with the true system locally evolves along a given trajectory, we computed representational similarity along an embedded trajectory of length 300. As outlined above, we use three anchor points to create a direct visualization of the relative latent spaces. Figure 3 illustrates the temporal evolution of alignment along a given trajectory. Consistent with our broader findings, forecasters within the same family tend to form similar temporal representations, whereas trans-

formers and ESNs display distinct alignment patterns—particularly in the Lorenz-63, double pendulum, and random skew systems.

**Latent stitching reveals family-specific representational compatibility**

Table 4 summarizes absolute and relative stitching losses on the Lorenz-63 dataset. Within model families—MLPs with MLPs and transformers with transformers—relative stitching outperforms absolute stitching. Transformer decoders act as strong universal decoders, achieving low losses even with absolute representations. However, relative stitching offers no benefit across families, as seen when mapping transformer representations to MLP decoders. RNNs were excluded because their reliance on hidden states makes both in-family and cross-family stitching incompatible under our current setup, leaving hidden-state stitching for future work. Overall, these results show that representational compatibility—and hence the ability to "stitch" encoders and decoders—is largely confined to model families that share similar latent geometries.

| enc/dec | MLP | | N-MLP | | K-MLP | | TF | | N-TF | | K-TF | |
|---|---|---|---|---|---|---|---|---|---|---|---|---|
| | Abs. | Rel. | Abs. | Rel. | Abs. | Rel. | Abs. | Rel. | Abs. | Rel. | Abs. | Rel. |
| MLP | 1.655 | **0.383** | 2.334 | **0.479** | **2.459** | 2.818 | **0.293** | 0.825 | 1.181 | **1.067** | **0.923** | 0.988 |
| N-MLP | 2.195 | **0.404** | 3.916 | **0.491** | 3.511 | **3.078** | **0.233** | 0.813 | **0.545** | 1.040 | 1.235 | **0.925** |
| K-MLP | 1.621 | **0.753** | 2.224 | **0.759** | 2.523 | **0.891** | **0.290** | 0.587 | 1.233 | **0.974** | 0.835 | **0.679** |
| TF | **1.538** | 2.019 | 2.389 | **1.754** | 2.118 | 9.517 | 0.265 | **0.043** | 1.383 | **0.587** | 0.599 | **0.076** |
| N-TF | **1.514** | 1.780 | 2.003 | **1.580** | 2.039 | 7.112 | 0.184 | **0.061** | 1.017 | **0.757** | 0.689 | **0.256** |
| K-TF | **1.590** | 2.011 | 2.095 | **1.750** | 2.129 | 9.466 | 0.254 | **0.042** | 1.325 | **0.586** | 0.840 | **0.075** |

Table 4: Cross-architecture average stitching loss (MSE) over encoder–decoder pairs for **absolute** (Abs.) and **relative** (Rel.) stitching. Each decoder column is independently normalized; darkest cell shows highest MSE and lightest shows lowest MSE respectively. Lower value per pair in bold.

**Similarity metrics and robustness.** Our primary measure of representational alignment is cosine similarity computed on anchor-based relative embeddings, which provides a geometry-agnostic comparison across architectures and seeds. To assess robustness to the choice of similarity metric, we additionally evaluate several standard representational similarity measures on the same models and dynamical systems. Specifically, we compare against representational similarity analysis (RSA), Procrustes-based alignment, and centered kernel alignment (CKA), each applied to the absolute latent representations following standard practice. Across all metrics, we observe consistent qualitative trends: in particular, strong within-family alignment for RNN- and MLP-based forecasters, systematically weaker alignment for transformers and ESNs, and a clear dissociation between forecasting accuracy and representational alignment for attention-based and reservoir models. While absolute similarity values vary across metrics, the family-level structure and relative ordering of architectures remain stable (Appendix Figure 15).

# 6 Discussion and Conclusions

This study examined neural forecasters for dynamical systems through the lens of *relative embeddings* (Moschella et al., 2023). Across periodic, quasi-periodic, and chaotic regimes, we observed reproducible, family-specific alignment patterns, alongside cases where strong forecasting performance coexisted with comparatively low cross-forecaster and system alignment—most notably in transformers and ESNs. These findings suggest that task loss alone does not fully capture how forecasters internalize latent geometry, echoing prior observations that similar task performance in neural networks can arise from distinct representational organizations (Kriegeskorte et al., 2008; Kornblith et al., 2019). Together, these results position representational alignment as a complementary dimension for understanding and evaluating neural forecasters.

To interpret these findings, we first clarify what relative alignment measures reveal about learned representations. Relative embeddings do not require estimating an explicit alignment or mapping between latent spaces. In contrast to Procrustes-based approaches, they do not assume linear or isometric correspondence

between representations (Gower, 1975; Schönemann, 1966). Unlike CKA, which compares representations through similarity matrices computed on matched inputs, relative embeddings define a shared relational coordinate system via similarities to a fixed set of anchors (Kornblith et al., 2019; Moschella et al., 2023). Alignment often increased with forecasting quality, but not universally. Model families with different inductive biases appear to summarize temporal context in distinct ways, achieving accurate predictions despite divergent relational geometries. In practice, representational alignment analysis thus complements forecasting loss functions when stability, interpretability, or transferability are priorities. Although not serving as evidence of learned physical dynamics [‡], it exposes architectural inductive biases and task-induced geometry.

To further interpret the observed family-level alignment patterns, it is instructive to relate them to architectural inductive biases within the shared encoder–propagator–decoder framework. RNN-based forecasters maintain a recursively updated hidden state, which induces temporally coherent latent-state evolution and results in consistently high representational alignment across seeds and architectural variants. MLP forecasters compress each input window into a single global latent representation via a fixed feedforward mapping, yielding a different, but relatively stable within-family geometry. In contrast, Transformer encoders construct token-wise contextual representations in parallel through self-attention, without architectural pressure to form smooth or trajectory-like latent representations. As a result, the representational regime for Transformers seems to focus on contextual summarization rather than latent-state evolution, which helps explain why strong forecasting accuracy can coexist with comparatively weaker geometric alignment. Finally, ESNs rely on fixed random reservoirs with only a trained readout layer, so reservoir trajectories primarily reflect internal reservoir dynamics rather than task-induced structure, accounting for their systematically lower alignment with the ground-truth relative representation.

A practical advantage of relative embeddings is more stable and interpretable visualization of learned latents. Standard projections of *absolute* embeddings (e.g., PCA) are sensitive to arbitrary rotations and scalings across seeds. Anchor-based relative spaces define a shared reference frame, making low-dimensional projections and neighborhood relations comparable across models. This facilitates diagnostics analogous to representational dissimilarity matrices in RSA (Kriegeskorte et al., 2008) and population "hyperalignment" in neuroimaging (Haxby et al., 2011). Combined with PCA, t-distributed Stochastic Neighbor Embedding (t-SNE) (van der Maaten & Hinton, 2008), or Uniform Manifold Approximation and Projection (UMAP) (McInnes et al., 2018) such spaces enable tracking of training trajectories, identifying attractor-specific regimes, and monitoring representational drift. In short, relative embeddings turn visualization from an exploratory tool into a quantitative diagnostic of representational geometry.

Alignment may serve several practical roles. It can serve as an auxiliary selection criterion during model development—favoring configurations that jointly achieve low forecast error and high representational agreement, particularly when downstream stitching or transfer is anticipated. Alignment trajectories during training may provide early warnings for overfitting or instability, for example when the relative space fragments. Finally, stitching encoders and decoders is more feasible when embeddings are computed relative to a common anchor set (Moschella et al., 2023). Together, these roles highlight alignment as a lightweight yet informative signal for model selection, monitoring, and interoperability.

Our analysis relies on a finite anchor set and a chosen similarity function. Too few anchors reduce discriminability; too many increase computational cost. While we empirically found stable behavior beyond a moderate anchor budget, adaptive anchor selection (e.g., farthest-point sampling or clustering) could improve robustness in higher dimensions. Relative embeddings are also less sensitive to certain non-isometric deformations (e.g., local shear). Complementary approaches based on geodesic/transport-aware comparisons—e.g., OT-based (optimal transport, OT) anchor bootstrapping of Cannistraci et al. (2023) and latent-space translation (Maiorca et al., 2023)—may capture finer structure. Finally, we focused on simulated benchmarks with controlled noise; assessments on high-dimensional and real-world systems will be necessary to test scalability and domain robustness. These limitations delineate a clear path toward more adaptive and geometry-aware alignment frameworks.

Several directions appear promising. (i) *Adaptive anchor selection* and bootstrapped ensembling of relative spaces could further stabilize estimates under limited data. In the context of dynamical systems, anchors

---

[‡]Preliminary experiments with a simple linear readout probe are shown in E

could be selected more informatively by targeting representative regions of the attractor or dynamically salient states. (ii) *Alignment-aware training*—for instance through auxiliary losses or early-stopping criteria—might promote generalizable latent representations. (iii) *Richer comparators*, including OT-based or spectral/functional-map techniques, could link alignment more tightly to long-horizon accuracy (García-Castellanos et al., 2024; Fumero et al., 2025). (iv) *Disentangling architectural and algorithmic effects*, for example by comparing standard training with long-horizon–aware objectives or alternative optimization schemes, may clarify which aspects of alignment are architecture-driven. (v) *Extended evaluations*, including long-term statistics, spectral properties, or topological features, could reveal whether alignment better predicts faithful dynamical behavior beyond short-horizon MSE. (vi) *Applications* to scientific forecasting and control may benefit from alignment-guided ensembling and forecaster monitoring. More broadly, integrating representational alignment into training and evaluation may help unify geometric, statistical, and dynamical perspectives on learning in neural systems.

Our findings show that neural forecasters develop reproducible, family-specific representational geometries that can diverge despite similar forecasting accuracy. This dissociation underscores the need for evaluation metrics that go beyond task performance and capture the geometry of learned latent geometry. By aligning latent spaces through anchor-based relative embeddings, we provide a simple and reproducible approach to study how different model families internalize structure in time-evolving systems. Relative geometry offers a compact, interpretable, and reproducible lens on learned representations—one that may help bridge analyses of artificial and biological neural systems.

## 7 Broader Impact Statement

This work is methodological and focuses on analyzing learned representations in neural forecasters rather than developing deployable prediction systems. In addition to canonical dynamical benchmarks, we include a preliminary analysis on open, de-identified intracranial EEG (iEEG) recordings, using the dataset solely as a testbed for representation analysis in a high-dimensional real-world setting. The study does not aim to perform clinical inference, diagnosis, or intervention, and all human-related data are observational and ethically released.

A key limitation is that representational alignment should not be interpreted as evidence of model correctness, causal validity, or recovery of true neural dynamics. Overall, the work presents low societal risk and contributes tools for understanding and comparing internal representations in neural and scientific time-series models beyond task performance alone.

## 8 Funding

Computational resources were provided by the Max Planck Computing and Data Facility (MPCDF). D.K. was supported by the DAAD project SECAI (project no. 57616814), funded by the German Federal Ministry of Research, Technology and Space (BMFTR). N.S. was supported by BMFTR through ACONITE (grant no. 16IS22065) and the Center for Scalable Data Analytics and Artificial Intelligence (ScaDS.AI) Leipzig, as well as by the European Union and the Free State of Saxony through BIOWIN.

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

# A    Remaining Experimental Results

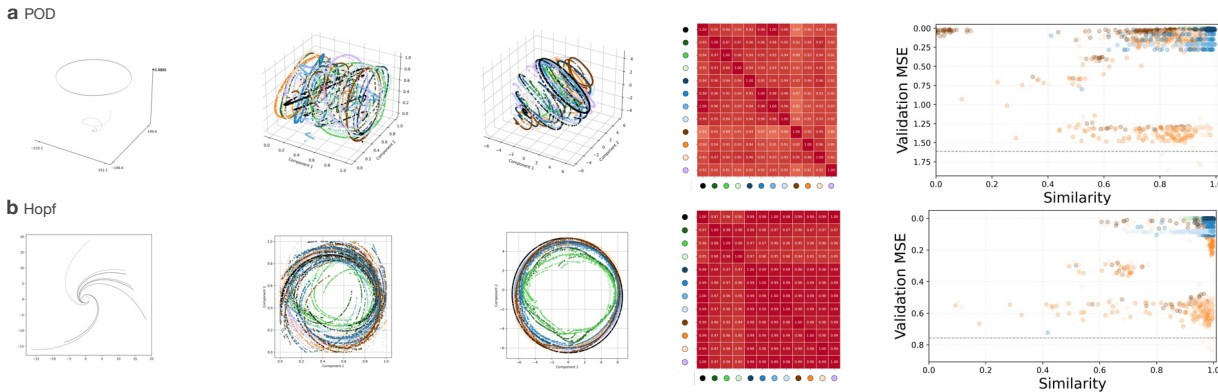

Figure 4: **Forecasting and representational alignment.** (a, b) Example systems: proper orthogonal decomposition (POD)-wake (a), Hopf (b). Columns show time series trajectories, absolute embeddings (visualized with principal component analysis (PCA); we plot the first 2 components for 2-dimensional systems and 3 components for the rest of the systems), relative embeddings (PCA), cross-forecaster similarity heatmaps (averaged over five seeds) with the order of True System, MLP, Koopman MLP, NODE MLP, RNN, Autoregressive RNN, Koopman RNN, NODE RNN, Transformer, NODE Transformer, Koopman Transformer, and ESN; and alignment–performance of forecasters with different hyperparameter settings.

Table 5: Results for `double_pendulum`

| Model | Performance (↓) | | | Similarity (↑) | | |
|---|---|---|---|---|---|---|
| | MSE | MAE | RMSE | Cosine | Top-1 | Spearman $\rho$ |
| MLP | $0.4122 \pm 0.0096$ | $0.4573 \pm 0.0049$ | $0.6420 \pm 0.0075$ | $0.6745 \pm 0.0105$ | $0.4262 \pm 0.0065$ | $0.6754 \pm 0.0115$ |
| Koopman MLP | $0.4314 \pm 0.0267$ | $0.4871 \pm 0.0202$ | $0.6565 \pm 0.0202$ | $0.6097 \pm 0.0241$ | $0.4112 \pm 0.0076$ | $0.6033 \pm 0.0287$ |
| NODE MLP | $0.3573 \pm 0.0102$ | $0.4364 \pm 0.0033$ | $0.5977 \pm 0.0085$ | $0.6212 \pm 0.0076$ | $0.4186 \pm 0.0025$ | $0.6211 \pm 0.0100$ |
| RNN | $0.2879 \pm 0.0117$ | $0.3618 \pm 0.0095$ | $0.5365 \pm 0.0109$ | $0.8254 \pm 0.0059$ | $0.5044 \pm 0.0151$ | $0.8197 \pm 0.0048$ |
| Autoregressive RNN | $0.3994 \pm 0.0494$ | $0.4450 \pm 0.0407$ | $0.6310 \pm 0.0397$ | $\mathbf{0.8791 \pm 0.0241}$ | $0.5526 \pm 0.0434$ | $\mathbf{0.8708 \pm 0.0256}$ |
| Koopman RNN | $0.4881 \pm 0.0843$ | $0.5148 \pm 0.0381$ | $0.6966 \pm 0.0590$ | $0.8608 \pm 0.1458$ | $\mathbf{0.5928 \pm 0.0279}$ | $0.8622 \pm 0.1272$ |
| NODE RNN | $0.3799 \pm 0.0332$ | $0.4512 \pm 0.0233$ | $0.6159 \pm 0.0270$ | $0.8631 \pm 0.0348$ | $0.5274 \pm 0.0210$ | $0.8528 \pm 0.0382$ |
| Transformer | $\mathbf{0.0072 \pm 0.0010}$ | $\mathbf{0.0685 \pm 0.0049}$ | $\mathbf{0.0846 \pm 0.0057}$ | $0.6468 \pm 0.1621$ | $0.4360 \pm 0.0150$ | $0.6485 \pm 0.1541$ |
| Koopman Transformer | $0.0129 \pm 0.0004$ | $0.0931 \pm 0.0023$ | $0.1135 \pm 0.0019$ | $0.7574 \pm 0.0240$ | $0.4462 \pm 0.0067$ | $0.7512 \pm 0.0245$ |
| NODE Transformer | $0.0178 \pm 0.0040$ | $0.1072 \pm 0.0098$ | $0.1329 \pm 0.0144$ | $0.3844 \pm 0.0465$ | $0.3986 \pm 0.0173$ | $0.4068 \pm 0.0378$ |
| ESN | $1.3081 \pm 0.0863$ | $0.7915 \pm 0.0342$ | $1.1432 \pm 0.0379$ | $0.1601 \pm 0.0014$ | $0.3808 \pm 0.0010$ | $0.1501 \pm 0.0015$ |

Table 6: Results for `random_skew`

| Model | Performance (↓) | | | Similarity (↑) | | |
|---|---|---|---|---|---|---|
| | MSE | MAE | RMSE | Cosine | Top-1 | Spearman $\rho$ |
| MLP | $0.9778 \pm 0.0532$ | $0.6652 \pm 0.0174$ | $0.9886 \pm 0.0271$ | $0.7533 \pm 0.0227$ | $0.8138 \pm 0.0034$ | $0.7724 \pm 0.0159$ |
| Koopman MLP | $0.8137 \pm 0.1571$ | $0.5918 \pm 0.0409$ | $0.8989 \pm 0.0837$ | $0.8523 \pm 0.0229$ | $0.8304 \pm 0.0038$ | $0.8440 \pm 0.0279$ |
| NODE MLP | $0.9145 \pm 0.0960$ | $0.6545 \pm 0.0195$ | $0.9552 \pm 0.0509$ | $0.7384 \pm 0.0174$ | $0.8090 \pm 0.0047$ | $0.7536 \pm 0.0159$ |
| RNN | $0.8815 \pm 0.0283$ | $0.6481 \pm 0.0081$ | $0.9388 \pm 0.0151$ | $0.8155 \pm 0.0204$ | $0.7840 \pm 0.0091$ | $0.8058 \pm 0.0209$ |
| Autoregressive RNN | $0.7683 \pm 0.0171$ | $0.5823 \pm 0.0089$ | $0.8765 \pm 0.0097$ | $0.8099 \pm 0.0153$ | $0.8178 \pm 0.0134$ | $0.8059 \pm 0.0153$ |
| Koopman RNN | $0.7356 \pm 0.0369$ | $0.5991 \pm 0.0196$ | $0.8575 \pm 0.0213$ | $0.8329 \pm 0.0406$ | $0.8394 \pm 0.0280$ | $0.8338 \pm 0.0362$ |
| NODE RNN | $0.7211 \pm 0.0172$ | $0.5897 \pm 0.0055$ | $0.8491 \pm 0.0102$ | $0.8617 \pm 0.0213$ | $\mathbf{0.8528 \pm 0.0202}$ | $0.8558 \pm 0.0189$ |
| Transformer | $\mathbf{0.4378 \pm 0.0074}$ | $0.4459 \pm 0.0083$ | $\mathbf{0.6617 \pm 0.0056}$ | $0.7417 \pm 0.0355$ | $0.8000 \pm 0.0158$ | $0.7426 \pm 0.0321$ |
| Koopman Transformer | $0.4690 \pm 0.0150$ | $0.4600 \pm 0.0073$ | $0.6848 \pm 0.0108$ | $0.7288 \pm 0.0211$ | $0.8174 \pm 0.0093$ | $0.7342 \pm 0.0195$ |
| NODE Transformer | $0.4429 \pm 0.0105$ | $\mathbf{0.4425 \pm 0.0121}$ | $0.6655 \pm 0.0079$ | $0.3904 \pm 0.0591$ | $0.5410 \pm 0.0433$ | $0.3957 \pm 0.0482$ |
| ESN | $0.6436 \pm 0.0436$ | $0.5803 \pm 0.0206$ | $0.8019 \pm 0.0268$ | $\mathbf{0.9074 \pm 0.0211}$ | $0.8340 \pm 0.0199$ | $\mathbf{0.9008 \pm 0.0185}$ |

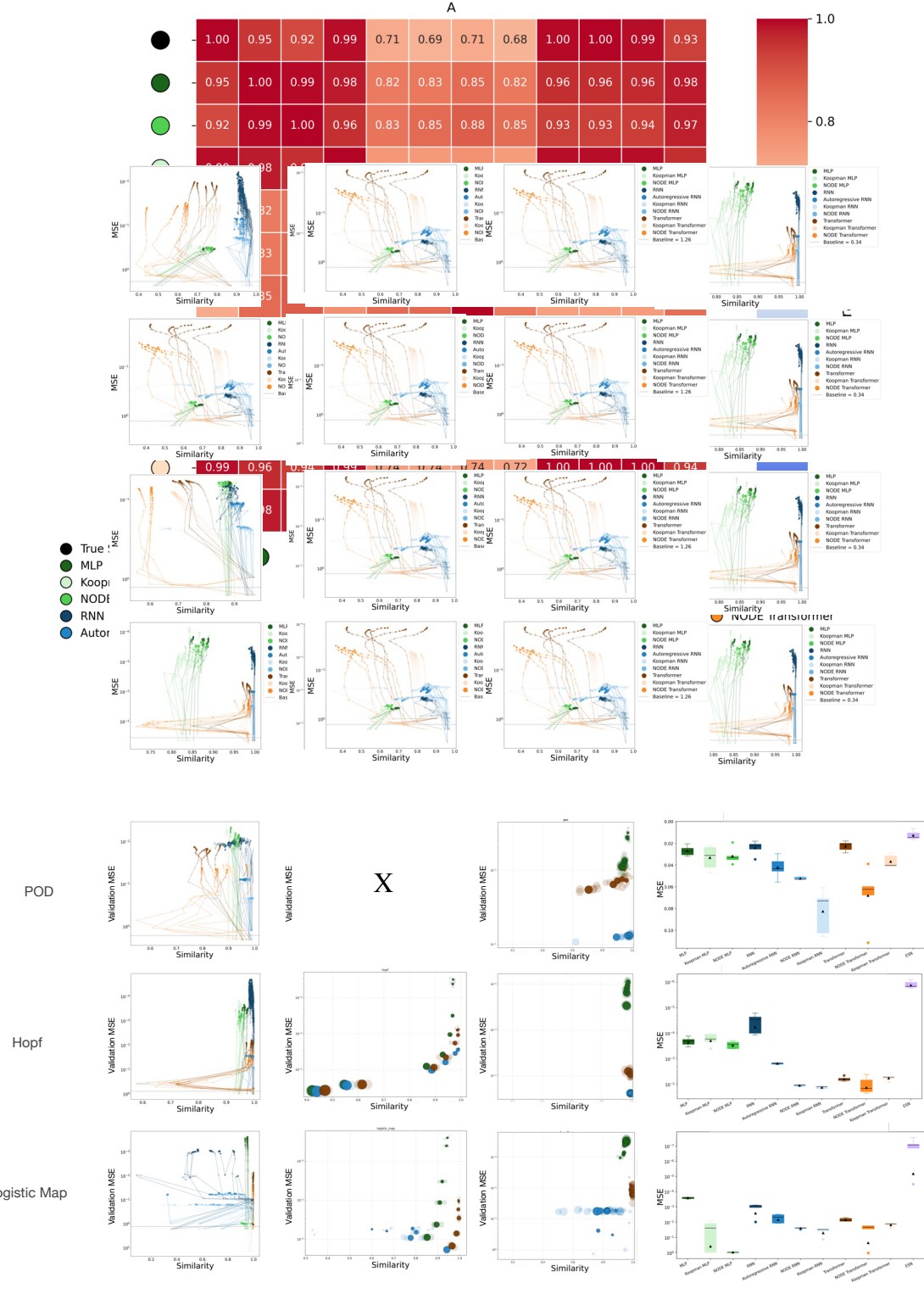

Figure 6: **Perturbation experiments for POD-wake, Hopf and Logistic Maps.** (X) The noise experiment could not be performed because the data were obtained from Brunton et al. (2016), and only the principal components—not the original data—were available.

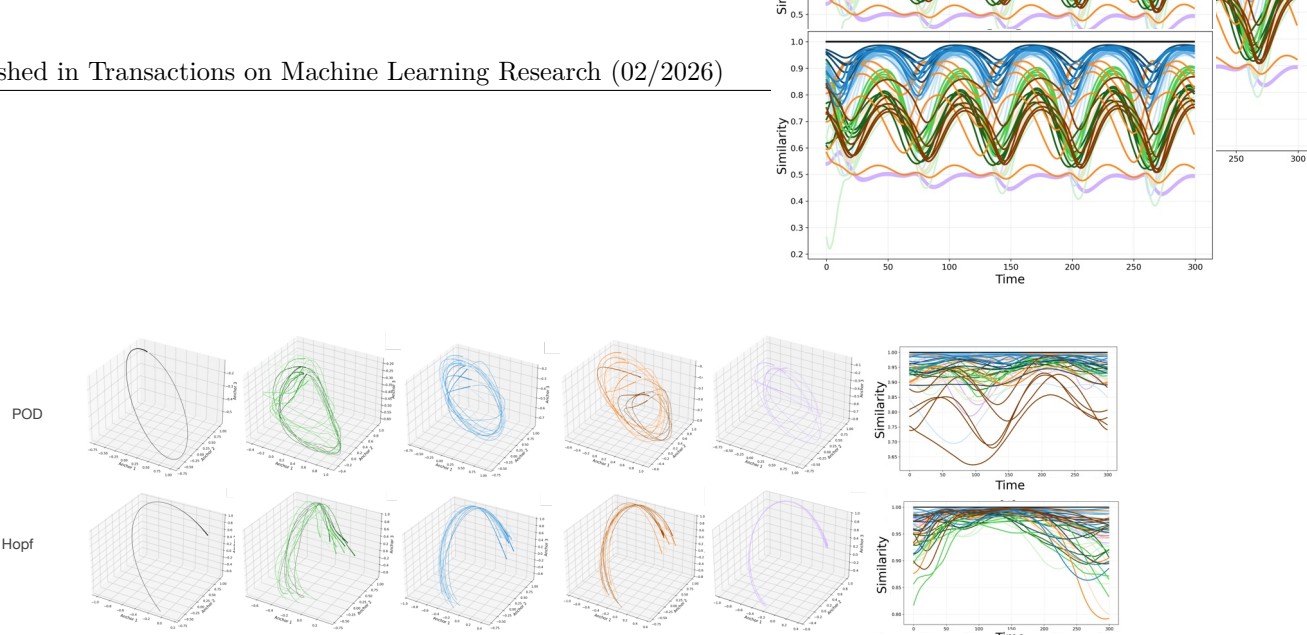

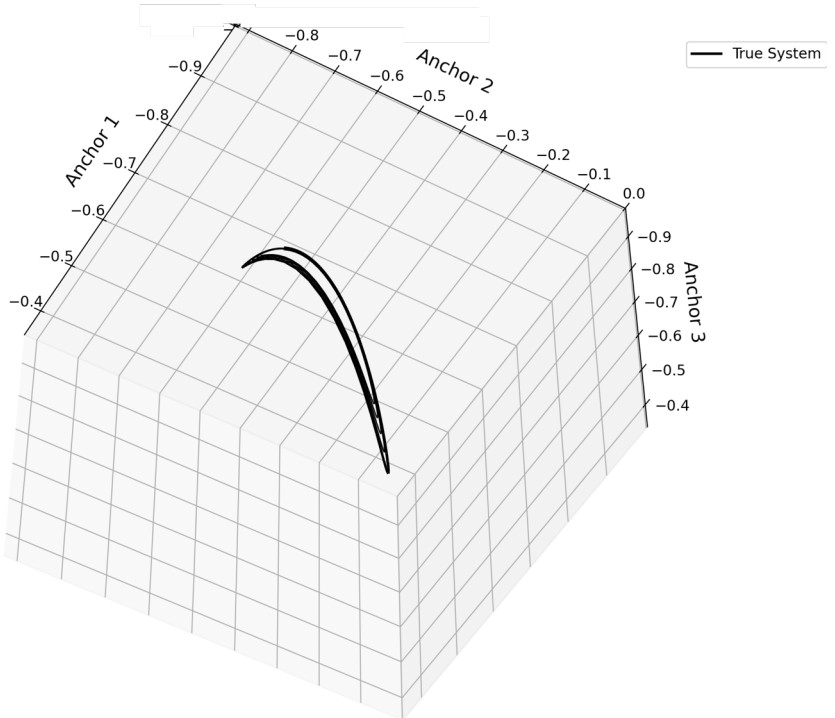

Figure 7: **Temporal alignment visualization for POD-wake and Hopf.**

Figure 8: **Temporal alignment rotated visualization of the True System of Lorenz.**

Table 7: Results for `spiral`

| | Performance (↓) | | | Similarity (↑) | | |
|---|---|---|---|---|---|---|
| Model | MSE | MAE | RMSE | Cosine | Top-1 | Spearman $\rho$ |
| MLP | **0.0002 ± 0.0000** | 0.0059 ± 0.0004 | **0.0128 ± 0.0010** | 0.8848 ± 0.0122 | 0.4020 ± 0.0029 | 0.8788 ± 0.0104 |
| Koopman MLP | 0.0002 ± 0.0001 | 0.0049 ± 0.0006 | 0.0136 ± 0.0035 | 0.8700 ± 0.0200 | 0.4200 ± 0.0035 | 0.8573 ± 0.0232 |
| NODE MLP | 0.0003 ± 0.0001 | 0.0067 ± 0.0012 | 0.0178 ± 0.0028 | 0.8796 ± 0.0127 | 0.4004 ± 0.0067 | 0.8698 ± 0.0121 |
| RNN | 0.0002 ± 0.0001 | 0.0108 ± 0.0017 | 0.0153 ± 0.0027 | 0.9917 ± 0.0039 | **0.6222 ± 0.0442** | 0.9906 ± 0.0038 |
| Autoregressive RNN | 0.0017 ± 0.0004 | 0.0298 ± 0.0044 | 0.0407 ± 0.0044 | 0.9936 ± 0.0040 | 0.5424 ± 0.0452 | 0.9921 ± 0.0033 |
| Koopman RNN | 0.0061 ± 0.0009 | 0.0633 ± 0.0044 | 0.0782 ± 0.0055 | **0.9952 ± 0.0033** | 0.5570 ± 0.0229 | **0.9938 ± 0.0026** |
| NODE RNN | 0.0043 ± 0.0006 | 0.0527 ± 0.0037 | 0.0658 ± 0.0045 | 0.9913 ± 0.0036 | 0.5330 ± 0.0208 | 0.9880 ± 0.0040 |
| Transformer | 0.0013 ± 0.0003 | 0.0254 ± 0.0033 | 0.0364 ± 0.0035 | 0.9822 ± 0.0040 | 0.4324 ± 0.0061 | 0.9774 ± 0.0037 |
| Koopman Transformer | 0.0025 ± 0.0009 | 0.0381 ± 0.0066 | 0.0493 ± 0.0095 | 0.9792 ± 0.0125 | 0.4344 ± 0.0074 | 0.9743 ± 0.0114 |
| NODE Transformer | 0.0044 ± 0.0009 | 0.0495 ± 0.0042 | 0.0664 ± 0.0068 | 0.9824 ± 0.0054 | 0.4216 ± 0.0105 | 0.9760 ± 0.0055 |
| ESN | 0.0003 ± 0.0001 | **0.0041 ± 0.0010** | 0.0151 ± 0.0053 | 0.9702 ± 0.0027 | 0.3874 ± 0.0058 | 0.9637 ± 0.0026 |

Table 8: Results for `pod`

| | Performance (↓) | | | Similarity (↑) | | |
|---|---|---|---|---|---|---|
| Model | MSE | MAE | RMSE | Cosine | Top-1 | Spearman $\rho$ |
| MLP | 0.0270 ± 0.0046 | 0.0862 ± 0.0071 | 0.1637 ± 0.0141 | 0.9511 ± 0.0107 | 0.8886 ± 0.0079 | 0.9024 ± 0.0175 |
| Koopman MLP | 0.0331 ± 0.0111 | 0.0931 ± 0.0125 | 0.1800 ± 0.0303 | 0.9277 ± 0.0179 | 0.8810 ± 0.0099 | 0.8679 ± 0.0308 |
| NODE MLP | 0.0317 ± 0.0075 | 0.0942 ± 0.0115 | 0.1770 ± 0.0227 | 0.9367 ± 0.0063 | 0.8758 ± 0.0118 | 0.8783 ± 0.0082 |
| RNN | 0.0242 ± 0.0063 | 0.0841 ± 0.0110 | 0.1547 ± 0.0195 | 0.9402 ± 0.0586 | 0.8876 ± 0.0558 | 0.9170 ± 0.0467 |
| Autoregressive RNN | 0.0422 ± 0.0097 | 0.1305 ± 0.0096 | 0.2043 ± 0.0239 | 0.9781 ± 0.0086 | 0.9328 ± 0.0186 | 0.9450 ± 0.0197 |
| Koopman RNN | 0.0826 ± 0.0201 | 0.2120 ± 0.0297 | 0.2857 ± 0.0347 | 0.9612 ± 0.0622 | 0.9280 ± 0.0457 | 0.9482 ± 0.0570 |
| NODE RNN | 0.0522 ± 0.0011 | 0.1697 ± 0.0019 | 0.2285 ± 0.0025 | **0.9953 ± 0.0022** | **0.9662 ± 0.0077** | **0.9741 ± 0.0137** |
| Transformer | 0.0227 ± 0.0043 | 0.0952 ± 0.0074 | 0.1501 ± 0.0142 | 0.8310 ± 0.0403 | 0.8506 ± 0.0176 | 0.8321 ± 0.0171 |
| Koopman Transformer | 0.0367 ± 0.0058 | 0.1310 ± 0.0097 | 0.1910 ± 0.0155 | 0.9149 ± 0.0101 | 0.8700 ± 0.0121 | 0.8767 ± 0.0135 |
| NODE Transformer | 0.0680 ± 0.0266 | 0.1645 ± 0.0173 | 0.2571 ± 0.0491 | 0.9019 ± 0.0490 | 0.8326 ± 0.0479 | 0.8574 ± 0.0668 |
| ESN | **0.0127 ± 0.0041** | **0.0483 ± 0.0076** | **0.1113 ± 0.0198** | 0.9262 ± 0.0210 | 0.8636 ± 0.0145 | 0.8774 ± 0.0384 |

Table 9: Results for `hopf`

| | Performance (↓) | | | Similarity (↑) | | |
|---|---|---|---|---|---|---|
| Model | MSE | MAE | RMSE | Cosine | Top-1 | Spearman $\rho$ |
| MLP | 0.0002 ± 0.0001 | 0.0080 ± 0.0006 | 0.0146 ± 0.0028 | 0.9672 ± 0.0088 | 0.5190 ± 0.0078 | 0.9596 ± 0.0119 |
| Koopman MLP | 0.0002 ± 0.0001 | 0.0061 ± 0.0006 | 0.0135 ± 0.0039 | 0.9463 ± 0.0119 | 0.5440 ± 0.0175 | 0.9300 ± 0.0227 |
| NODE MLP | 0.0003 ± 0.0001 | 0.0096 ± 0.0007 | 0.0169 ± 0.0027 | 0.9550 ± 0.0082 | 0.5160 ± 0.0162 | 0.9463 ± 0.0107 |
| RNN | 0.0001 ± 0.0000 | 0.0042 ± 0.0022 | 0.0070 ± 0.0031 | 0.9907 ± 0.0040 | 0.5474 ± 0.0297 | 0.9905 ± 0.0044 |
| Autoregressive RNN | 0.0015 ± 0.0001 | 0.0332 ± 0.0009 | 0.0386 ± 0.0012 | 0.9888 ± 0.0068 | 0.5638 ± 0.0358 | 0.9904 ± 0.0052 |
| Koopman RNN | 0.0130 ± 0.0016 | 0.1010 ± 0.0060 | 0.1137 ± 0.0069 | 0.9938 ± 0.0020 | 0.5976 ± 0.0284 | 0.9937 ± 0.0020 |
| NODE RNN | 0.0108 ± 0.0006 | 0.0919 ± 0.0026 | 0.1038 ± 0.0029 | 0.9963 ± 0.0014 | **0.6192 ± 0.0310** | **0.9958 ± 0.0014** |
| Transformer | 0.0061 ± 0.0012 | 0.0673 ± 0.0064 | 0.0776 ± 0.0076 | 0.9874 ± 0.0093 | 0.5452 ± 0.0218 | 0.9871 ± 0.0074 |
| Koopman Transformer | 0.0056 ± 0.0011 | 0.0645 ± 0.0044 | 0.0748 ± 0.0069 | 0.9895 ± 0.0064 | 0.5606 ± 0.0200 | 0.9891 ± 0.0061 |
| NODE Transformer | 0.0130 ± 0.0075 | 0.0893 ± 0.0290 | 0.1095 ± 0.0355 | 0.9901 ± 0.0061 | 0.5486 ± 0.0239 | 0.9883 ± 0.0062 |
| ESN | **0.0000 ± 0.0000** | **0.0004 ± 0.0000** | **0.0011 ± 0.0002** | **0.9978 ± 0.0003** | 0.5216 ± 0.0036 | 0.9949 ± 0.0005 |

Table 10: Results for `logistic_map`

| | Performance (↓) | | | Similarity (↑) | | |
|---|---|---|---|---|---|---|
| Model | MSE | MAE | RMSE | Cosine | Top-1 | Spearman $\rho$ |
| MLP | 0.0002 ± 0.0000 | 0.0118 ± 0.0006 | 0.0157 ± 0.0008 | 0.9489 ± 0.0074 | 0.0232 ± 0.0040 | 0.7468 ± 0.0035 |
| Koopman MLP | 0.3713 ± 0.4895 | 0.4102 ± 0.4452 | 0.4505 ± 0.4586 | 0.9906 ± 0.0047 | 0.0232 ± 0.0051 | **0.7468 ± 0.0044** |
| NODE MLP | 0.8986 ± 0.0191 | 0.8939 ± 0.0109 | 0.9479 ± 0.0101 | 0.9165 ± 0.0085 | 0.0228 ± 0.0046 | 0.7446 ± 0.0044 |
| RNN | 0.0025 ± 0.0038 | 0.0325 ± 0.0245 | 0.0421 ± 0.0302 | 0.7056 ± 0.1526 | **0.0302 ± 0.0059** | 0.6182 ± 0.1131 |
| Autoregressive RNN | 0.0065 ± 0.0041 | 0.0648 ± 0.0197 | 0.0771 ± 0.0255 | 0.6875 ± 0.2004 | 0.0240 ± 0.0074 | 0.5479 ± 0.2504 |
| Koopman RNN | 0.0483 ± 0.0427 | 0.1801 ± 0.0573 | 0.2074 ± 0.0817 | 0.6835 ± 0.2288 | 0.0264 ± 0.0054 | 0.5198 ± 0.2503 |
| NODE RNN | 0.0239 ± 0.0029 | 0.1429 ± 0.0062 | 0.1545 ± 0.0091 | 0.7063 ± 0.0921 | 0.0198 ± 0.0057 | 0.6261 ± 0.1181 |
| Transformer | 0.0068 ± 0.0015 | 0.0667 ± 0.0085 | 0.0820 ± 0.0092 | 0.9953 ± 0.0023 | 0.0230 ± 0.0074 | 0.7413 ± 0.0013 |
| Koopman Transformer | 0.0142 ± 0.0043 | 0.1008 ± 0.0098 | 0.1183 ± 0.0168 | 0.9908 ± 0.0023 | 0.0188 ± 0.0034 | 0.7398 ± 0.0010 |
| NODE Transformer | 0.2153 ± 0.4374 | 0.2831 ± 0.3738 | 0.3117 ± 0.3844 | **0.9964 ± 0.0008** | 0.0208 ± 0.0077 | 0.7413 ± 0.0019 |
| ESN | **0.0000 ± 0.0000** | **0.0007 ± 0.0011** | **0.0013 ± 0.0024** | 0.9307 ± 0.0083 | 0.0162 ± 0.0004 | 0.7401 ± 0.0008 |

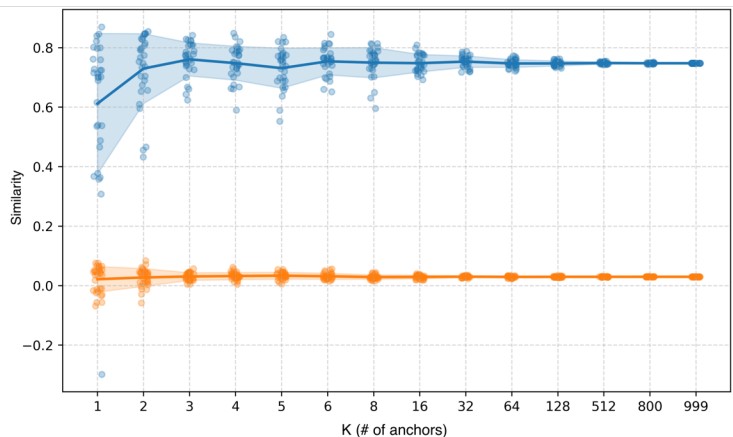

Figure 9: **Anchor ablation and baseline.** (Blue) Alignment vs. number of anchors $K$; lines show mean over 30 repeats. Stabilization occurs for $K \geq 16$; we choose $K = 80$ (vertical marker) for the main experiments. (Orange) Random baseline with disjoint anchor sets across spaces, yielding near-zero alignment.

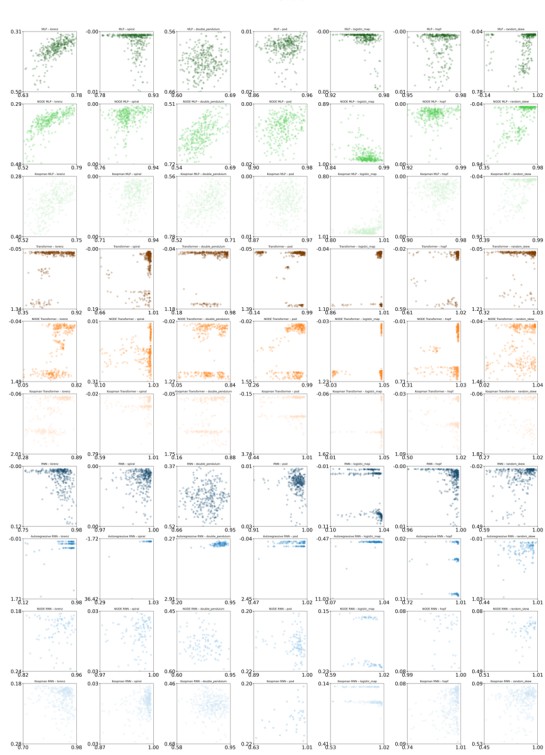

Figure 10: **Hyperparameter Tuning for All Systems and Models.** ESN is exluded since it needed additional manual hyperparameter tuning due to its sensitivity to hyperparameters and unstable nature.

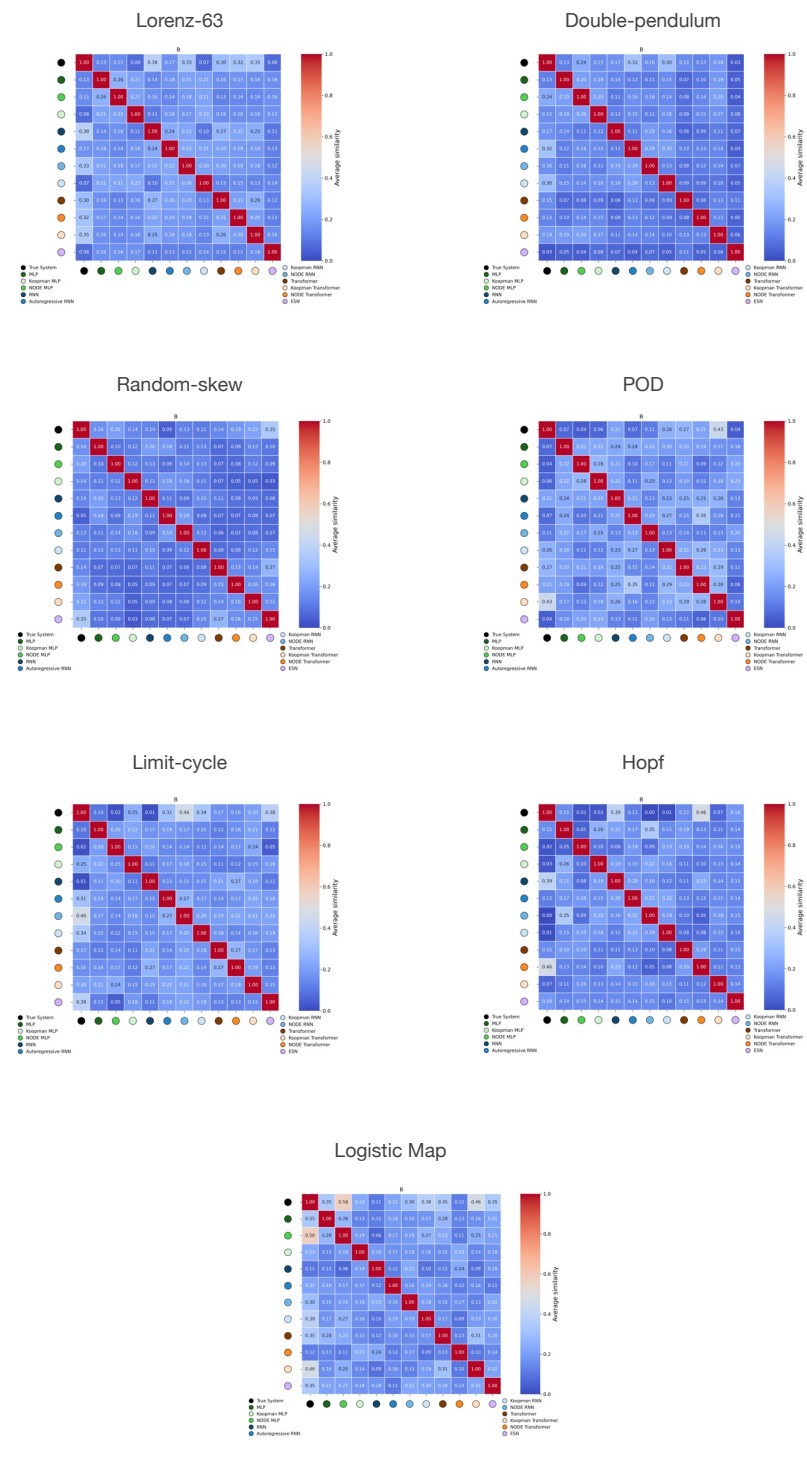

Figure 11: **Cross-model similarity using absolute embeddings.**

## B   Similarity Metrics

**T1 similarity.**   The T1 score measures the agreement in the identity of the most similar anchor across encoders. For each latent $\mathbf{z} \in \mathcal{V}$, we check whether the anchor with highest similarity under encoder 1 coincides with that under encoder 2:

$$\alpha_{\mathrm{T1}}\Big(\phi^{(1)}_{\theta^{(1)}_e}, \phi^{(2)}_{\theta^{(2)}_e}; \mathcal{V}\Big) = \frac{1}{|\mathcal{V}|} \sum_{\mathbf{z} \in \mathcal{V}} \mathbf{1}\Big[\arg\max_i \mathbf{r}^{(1)}_{\mathrm{rel},i}(\mathbf{z}) = \arg\max_i \mathbf{r}^{(2)}_{\mathrm{rel},i}(\mathbf{z})\Big].$$

**Rank similarity.**   The rank similarity evaluates how similarly two encoders order the set of anchors for each latent $\mathbf{z}$. For each encoder $s \in \{1, 2\}$, let $\mathrm{rank}_\downarrow(\mathbf{r}^{(s)}_{\mathrm{rel}}(\mathbf{z}))$ denote the vector of descending ranks (with 1 assigned to the largest component) of the relative similarity vector $\mathbf{r}^{(s)}_{\mathrm{rel}}(\mathbf{z})$, with ties resolved using a stable sort order. The average Spearman correlation between these rank vectors defines the rank similarity:

$$\alpha_{\mathrm{rank}}\Big(\phi^{(1)}_{\theta^{(1)}_e}, \phi^{(2)}_{\theta^{(2)}_e}; \mathcal{V}\Big) = \frac{1}{|\mathcal{V}|} \sum_{\mathbf{z} \in \mathcal{V}} \rho\Big(\mathrm{rank}_\downarrow\big(\mathbf{r}^{(1)}_{\mathrm{rel}}(\mathbf{z})\big), \mathrm{rank}_\downarrow\big(\mathbf{r}^{(2)}_{\mathrm{rel}}(\mathbf{z})\big)\Big),$$

where $\rho$ denotes Spearman's rank correlation coefficient, implemented as the Pearson correlation between the rank-transformed relative similarity vectors.

## C   Stitching Details

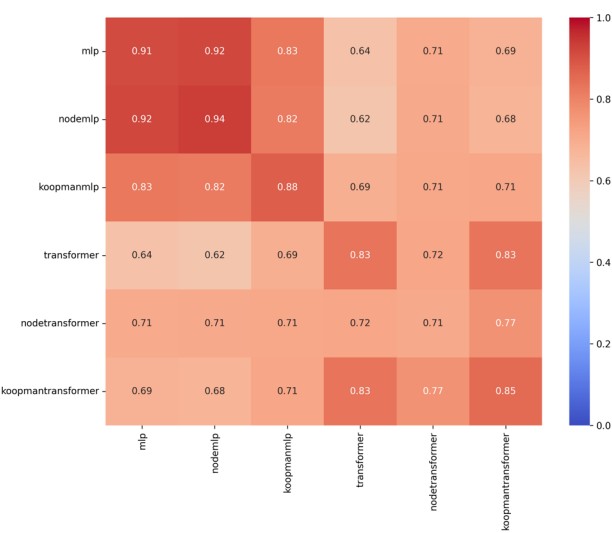

Figure 12:   Average cosine similarity over all 5 seeds for each forecaster pair, excluding exact same forecaster and seed. Models were trained on relative latent spaces rather than absolute spaces.

# D    Dynamical Systems Details

**Lorenz–63 (3-D chaotic ODE).** $\dot{x} = \sigma(y-x)$, $\dot{y} = x(\rho-z)-y$, $\dot{z} = xy-\beta z$, with $\sigma = 10$, $\rho = 28$, $\beta = 8/3$. Initial states are sampled from $[-20, 20]^3$ and integrated with Runge–Kutta 45 (RK45) at $\Delta t = 0.01$. Its compact phase space and positive Lyapunov exponent ($\approx 0.91$) make it a classical multi-step-forecast benchmark.

**Stable limit cycle (2-D radial–spiral ODE).** $\dot{r} = \mu(R - r)$, $\dot{\theta} = \omega$, $(x,y) = (r\cos\theta, r\sin\theta)$, with $\mu = 1$, $R = 1$, $\omega = 1$. Trajectories start from $r_0 \sim \mathcal{U}[0, 20]$ and $\theta_0 \sim \mathcal{U}[0, 2\pi]$; integration uses RK45 with $\Delta t = 0.01$.

**Double pendulum (4-D Hamiltonian chaos).** Two unit–mass, unit–length links evolve under gravity $g{=}9.81$. Writing the state as $(\theta_1, \theta_2, \omega_1, \omega_2)$ with $\Delta = \theta_2 - \theta_1$, the equations of motion are

$$\dot{\theta}_1 = \omega_1, \qquad \dot{\theta}_2 = \omega_2,$$

$$\dot{\omega}_1 = \frac{\omega_1^2 \sin\Delta \cos\Delta + g\sin\theta_2 \cos\Delta + \omega_2^2 \sin\Delta - 2g\sin\theta_1}{2 - \cos^2\Delta},$$

$$\dot{\omega}_2 = \frac{-\omega_2^2 \sin\Delta \cos\Delta + 2g\sin\theta_1 \cos\Delta - 2\omega_1^2 \sin\Delta - 2g\sin\theta_2}{2 - \cos^2\Delta}.$$

Initial angles are sampled from $[-20°, 20°]$ and angular velocities from $[-1, 1]$. Trajectories are integrated with RK45 at $\Delta t{=}0.01$. The system exhibits strongly chaotic, nearly energy–conserving motion, with a positive Lyapunov exponent of $\approx 1.5$.

**Hopf normal form (2-D near-critical oscillation).** $\dot{x} = \mu x - \omega y - (x^2+y^2)x$, $\dot{y} = \omega x + \mu y - (x^2+y^2)y$, with $\mu = 0$, $\omega = 1$. Starting points $(x_0, y_0) \sim \mathcal{U}[-2, 2]^2$ spiral onto a unit-radius limit cycle; $\Delta t = 0.01$ with RK45.

**Logistic map (1-D near-onset discrete chaos).** $x_{t+1} = 3.57\, x_t(1 - x_t)$ with $x_0 \sim \mathcal{U}(0, 1)$; sequences of length $T{=}500$ are recorded at an effective step $\Delta t = 0.1$.

**Fluid wake behind a cylinder (POD-wake coefficients; $d = 3$).** We adopt the three leading Proper-Orthogonal-Decomposition coefficients from Brunton et al. (2016) (Re = 100, Strouhal $\approx 0.16$). We supply 10 trajectories per split, each of $T{=}500$ snapshots sampled at $\Delta t = 0.2$; only z-score normalisation is applied.

**Skew-product of 3-D chaotic founders (6-D weakly coupled ODE).** Following Lai et al. (2025), select two founders from {Lorenz–63, Rössler, Chen}, jitter parameters by multiplicative log-normal noise ($\log s \sim \mathcal{N}(0, 0.15^2)$, sign preserved), and couple them in a skew-product: the first 3-D system $x \in \mathbb{R}^3$ drives the second $y \in \mathbb{R}^3$ via a weak injection into the first response coordinate. Writing $\dot{x} = f_a(x; p_a)$ and $\dot{y} = f_b(y; p_b)$ for the founders with jittered parameters,

$$\dot{x} = f_a(x; p_a), \qquad \dot{y} = f_b(y; p_b) + \varepsilon\, e_1\, x_1, \quad \varepsilon = 0.05,\ e_1 = (1, 0, 0)^\top.$$

Founder templates and nominal seeds:

$$\text{Lorenz–63: } \dot{x} = \sigma(y-x),\ \dot{y} = x(\rho-z)-y,\ \dot{z} = xy-\beta z;\ (\sigma, \rho, \beta) = (10, 28, 8/3),\ x_0 = (1, 1, 1),$$
$$\text{Rössler: } \dot{x} = -y-z,\ \dot{y} = x+ay,\ \dot{z} = b+z(x-c);\ (a, b, c) = (0.2, 0.2, 5.7),\ x_0 = (0.1, 0, 0),$$
$$\text{Chen: } \dot{x} = a(y-x),\ \dot{y} = (c-a)x-xz+cy,\ \dot{z} = xy-bz;\ (a, b, c) = (35, 3, 28),\ x_0 = (-10, 0, 37).$$

A single skew system is sampled once per dataset; train/val/test splits then differ only by initial conditions. Initial states jitter the concatenated founder seeds $z_0 = [x_0; y_0]$ with i.i.d. Gaussian noise of scale 0.1. Trajectories are integrated with DOP853 at the dataset step $\Delta t$ (absolute tolerance $10^{-8}$, relative $10^{-6}$). We discard an initial warm-up fraction (default 10%) and keep the next $T$ steps. Runs are rejected if any state is non-finite, the radius exceeds $10^6$, or the summed channel variance falls below $10^{-6}$; on rejection we resample once.

# E  Probing state information in latent representations.

To assess the extent to which learned latent representations preserve information about the underlying system state, we perform a simple linear probing analysis on a representative dynamical system (Lorenz–63). For each trained model, we freeze the encoder and fit a single linear ridge regressor to decode the current observable state $x(t)$ from the corresponding latent representation $z(t)$, training on the training split and evaluating on held-out test data. When probing absolute latent representations, decoding performance is near-perfect across all model families, indicating that the encoder latents retain almost complete information about the instantaneous system state. When applying the same probe to anchor-based relative embeddings, decoding performance remains high but exhibits a modest, architecture-dependent reduction. This behavior is expected, as relative embeddings are designed to quotient out certain geometric degrees of freedom in order to enable cross-model comparison, rather than to preserve all linearly decodable structure. We emphasize that this probing analysis is intended as a sanity check on representational content rather than as evidence of system identification or recovery of governing dynamics. We report these results in Figure 13 and interpreted as bounding the information retained by the representations, rather than as a claim about learning the true dynamical system.

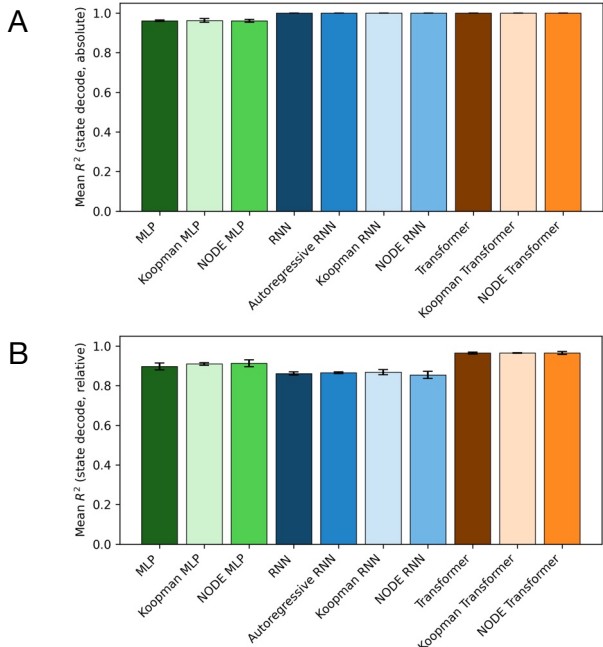

Figure 13: Mean test-set $R^2$ of a linear ridge probe decoding the current system state $x(t)$ from absolute latent representations $z(t)$ for the Lorenz–63 system, averaged over three random seeds (error bars denote standard deviation).

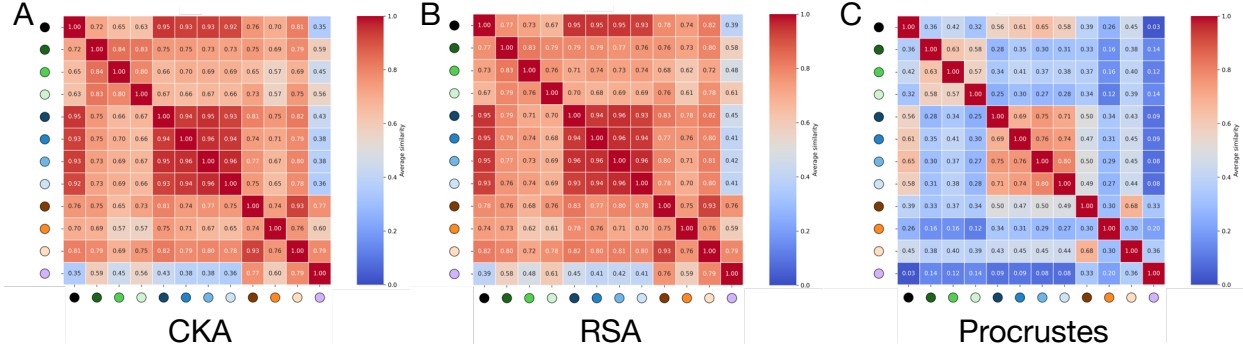

Figure 14: Cross model alignment using centered kernel alignment (CKA), representational similarity analysis (RSA), and Procrustes-based alignment. Values indicate the average of three seeds.

## F   Preliminary Results on iEEG Data.

We include this experiment as a preliminary external validation of the representational alignment analysis in a high-dimensional, real-world setting, using human intracranial EEG recordings from the first participant (ID1) of the SWEC–ETHZ dataset (Ghosh (2024)). In contrast to the simulated systems, where representational similarity can be measured directly against known ground-truth dynamics, such a comparison is not possible in the iEEG setting, as the underlying generative system is unknown.

The goal of this experiment is not to benchmark forecasting accuracy or to draw neuroscientific conclusions, but to assess whether the family-level alignment patterns observed in controlled dynamical systems persist when models are trained on real neural time series under otherwise comparable experimental conditions.

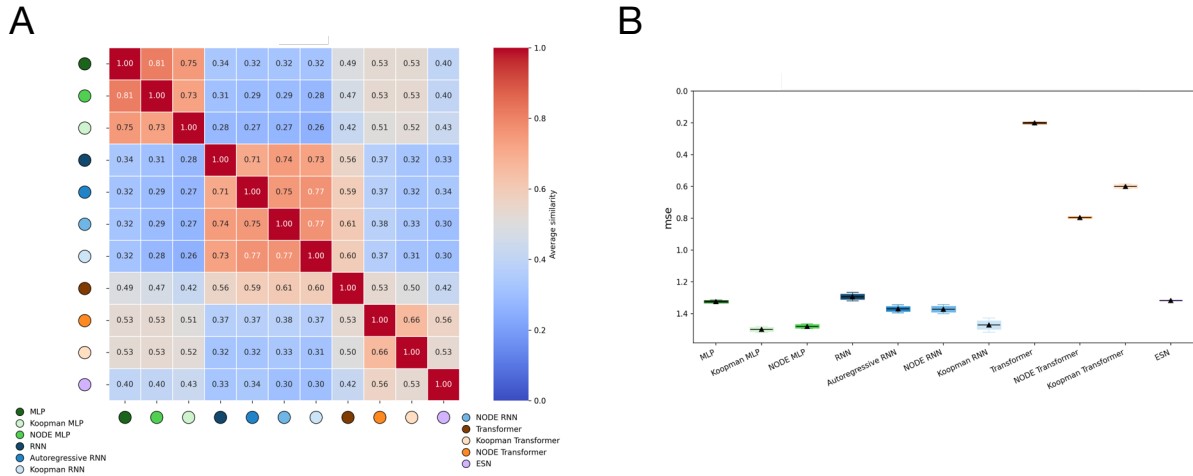

Figure 15: A. Cross model alignment in forecasters trained on forecasting iEEG dataset. B. Performances of forecasters.

