# OpenReview forum: "Relative Geometry of Neural Forecasters: Linking Accuracy and Alignment in Learned Latent Geometry"
_TMLR — Accepted by TMLR_

### Review · Reviewer_yRE7 · 2025-11-16

**Summary Of Contributions:**

**Summary:**

This paper conducts an empirical study of representational alignment in neural forecasting models applied to synthetic dynamical systems (e.g., Lorenz-63, double pendulum, skew-product maps). It adapts previously proposed anchor-based relative representation methods to compare latent spaces of different architectures. The authors report structure in representational similarity within and across model families and present preliminary evidence of encoder–decoder stitching across architectures.

---

**Key strengths:**

- Systematic empirical study across multiple dynamical systems and architecture families; well-executed evaluation methodology.
- Clear experimental framework and high reproducibility (simulation code, hyperparameters, and implementation details are provided).
- Some insightful findings, such as divergence between representational alignment and forecasting performance for architectures like transformers and ESNs, and structure in family-wise alignment.

---

**Key weaknesses**

- The learning objective shaping the latent representations is not clearly articulated. It is unclear why optimizing for forecasting loss should be expected to yield latent variables that reflect underlying system dynamics rather than generic short-horizon predictive encodings.
- Some claims risk overclaim: to my understanding, use of terms such as "learned dynamics" or "training dynamics" suggests recovery of true dynamical structure, but the experiments do not provide evidence to support that interpretation.
- Empirical evaluation focuses on low-dimensional synthetic systems, limiting the ability to assess generalization to real-world or higher-complexity settings.
- The anchor-based representational comparison closely follows prior work, with modest methodological extension beyond applying the technique to a new domain (though the authors clearly acknowledge this prior work).

**Additional Comments:**

The figures are strong and contribute meaningfully to the interpretability of the results. The work has the potential to serve as a reproducible empirical benchmark for studying representational alignment in neural forecasting models, particularly if the authors expand the experimental coverage and provide a more comprehensive and carefully packaged release of code, data, and evaluation tools.

**Audience:**

Yes

**Audience Explanation:**

The topic aligns with areas including representation learning, neural forecasting, and latent space analysis. Although the contribution is modest, the insights on representational alignment across forecasting architectures are likely to benefit some readers.

**Broader Impact Concerns:**

The work uses fully synthetic data and focuses on representation analysis. No significant ethical concerns are identified.

**Claims And Evidence:**

Yes

**Claims Explanation:**

The evidence is convincing with respect to representational alignment and forecasting comparisons. However, it does not adequately support claims that the models learn or recover underlying system dynamics. Without theoretical grounding or diagnostics linking latent structure to true dynamics, some interpretations appear overstated relative to the results.

**Requested Changes:**

I would suggest the authors consider the following revisions:

- Clarify the contribution relative to prior anchor-based representation alignment and stitching work (e.g., Moschella et al., 2023).
Please articulate which components are reused, which are adapted to the dynamical forecasting setting, and what new insights the present work contributes. As written, the methodological novelty is unclear.

- Clarify the learning objective and its connection to "dynamics".
State explicitly whether the goal is: (a) forecasting performance only, (b) learning representations that reflect underlying system dynamics, or (c) both. If (b) or (c), provide theoretical or empirical justification (e.g., diagnostics showing that latent spaces preserve dynamical invariants or structure) rather than assuming forecasting loss implies learning dynamics.

- Expand evaluation beyond low-dimensional synthetic systems.
Including at least one higher-dimensional or real-world dataset (e.g., realistic fluid/CFD systems, spatiotemporal climate/traffic/energy forecasting) would improve external validity and strengthen claims about the generality of the findings.

---

> ### Author Response · Authors · 2025-12-17
> **Scope, contribution, and interpretation of representational alignment**
>
> We thank the reviewer for the thoughtful and constructive feedback, and for recognizing the clarity, reproducibility, and systematic nature of the experimental study. We address the main concerns below.
>
> **Contribution relative to prior work.**
>
> We agree that the alignment method itself is not novel. In the revision, we will more clearly state that we reuse the anchor-based relative representation framework of Moschella et al. (2023). Our contribution is empirical: adapting this framework to encoder–propagator–decoder forecasters, applying it to dynamical systems forecasting, analyzing family-level representational structure and alignment–performance trade-offs, and evaluating latent stitching in this setting.
>
> **Terminology and scope (“learned dynamics”).**
>
> We agree that some terminology in the manuscript risks overstating our claims. We do not intend to suggest recovery of true physical state variables, governing equations, or dynamical invariants. In the revision, we will replace “learned dynamics” with “learned latent geometry”. The term “training dynamics” is a misnomer, we mean to refer to the development of prediction accuracy and geometric alignment over training. We will change Fig. 2 labels accordingly. We will explicitly clarify in the Introduction and Discussion that our goal is to study representational geometry induced by forecasting objectives, not system identification or recovery of ground-truth dynamics. We leave this direction for future work.
>
> **Learning objective and relation to dynamics.**
>
> Our models are trained purely to minimise forecasting loss. We will clarify that representational alignment is used as an analysis tool, not as an assumed consequence of the objective. The observed alignment (and misalignment) with the true system reflects architectural inductive biases and task-induced representations, rather than evidence that forecasting loss implies learning the underlying dynamics. We will strengthen this clarification and explicitly frame the divergence between accuracy and alignment (e.g., in transformers and ESNs) as a key empirical finding.
>
> **Scope of evaluation.**
>
> We acknowledge the limitation of focusing on low-dimensional synthetic systems. These were chosen for control and interpretability, which we believe is appropriate for a first systematic study of representational alignment across forecasting architectures. We agree that extending the analysis to higher-dimensional or real-world systems is an important next step. In the revision, we will strengthen the discussion of this limitation and clarify the regimes in which our conclusions are intended to apply. Where feasible, we will also explore extending the empirical evaluation to a higher-dimensional simulated system or real-world dataset.
>
>
> We appreciate the reviewer’s suggestions, which will substantially improve the clarity, positioning, and scope of the paper.

---

> > ### Comment · Reviewer_yRE7 · 2025-12-17
> >
> > Thank you for your response. If the revisions are implemented faithfully as described, my concerns will be resolved. Nonetheless, I recommend the inclusion of at least one additional empirical validation in a more practical setting to strengthen the paper’s impact.

---

### Review · Reviewer_pBzn · 2025-12-03

**Summary Of Contributions:**

Neural forecasters (MLPs, RNNs, Transformers, and ESNs) take a window of past states and predict future states, but their internal hidden-states or latent representations evolve differently w.r.t the architecture. This paper examines how these internal latent trajectories reflect the underlying dynamical system, and how similar the internal latent representations across different neural forecasters. To achieve this, the authors select the subject of anchor-based relative embeddings: they select a shared set of anchor points from the dataset and represent each model’s latent vectors by their similarities to these anchors, producing geometry-agnostic representations that can be compared across architectures. The evaluation focuses on periodic, quasi-periodic, and chaotic systems. The authors observe that both RNNs and MLPs learn similar latent geometry within their respective families even under different seeds, while ESNs and Transformers show weaker and less stable alignment. Across architectures, they observe that Transformers (and sometimes ESNs) result in higher prediction accuracy even when their alignment with the underlying dynamics is low, indicating a disconnection between forecasting performance and representational similarity. Further, the authors interpret that the temporal evaluation of alignment is shown to be sensitive to noise and input-sequence length, with effects varying across neural forecasters.

**Contributions:**
1. *Introduction of anchor-based relative embeddings to neural forecasters*: This study adapts the relative-embedding alignment approach to neural forecasters, making them geometric-agnostic and directly comparable across different architectures.
2. *Uncovering the alignment patterns across architectures*: This study interprets how representational alignment relates to forecasting accuracy across architectures and provides architecture-specific behaviors.
3. *Comprehensive evaluation*: The study presents an extensive assessment of seven canonical systems including, continuous, discrete, periodic, quasi-periodic, and chaotic, and four different architectures. Further, the authors compute the underlying latent alignment similarity within their model families, cross-modal families, and perform robustness studies including noise input and longer sequence lengths.

**Technical summary:**
This is primarily an empirical study, and its methodology involves the following components:
1. *Representational alignment framework:* The authors present the alignment framework with components including data (trajectory of data from seven canonical systems), neural forecasters (model architecture), encoder’s latent representation (applying neural forecaster on input windows), anchor-based relative embeddings (obtained from a fixed subset), and similarity metrics (cosine, T1 and rank similarity) between two embeddings is well explained.
2. *Neural forecasters and propagated dynamics:* The authors present encoder -> propagator -> decoder architecture: (i) Encoder takes a window of past states and produces a latent vector, (ii) Evolves (or “propagates”) this latent state forward for H forecasting steps (z0​->z1​->z2​->....->zH​) and (iii) Decoder converts the final latent state zH​ into future predictions. Further, the authors discuss three types of propagators: (i) Identity, used in standard MLP, RNN and Transformer models, (ii) Neural ODE, used for continuous-latent dynamics, and (iii) Koopman, which enforces linear latent-time evolution.

**Experimental design/evaluation:**
The authors focus on (i) cross-family alignment, (ii) alignment and its relation to forecasting accuracy, and (iii) perturbation experiments.
1. *Cross-family alignment*: This analysis evaluates the representational alignment using anchor-based relative embeddings and measures the similarity between architectures by making them geometry-agnostic. Across seven canonical systems and four different architectures, this analysis provides cross-forecaster alignment beyond what can be observed in the original latent spaces.
2. *Alignment relation to forecasting accuracy*: This analysis provides whether the prediction accuracy of a true system relates to its underlying latent representation changes and how this relation varies across different neural forecasters. Specifically, the authors present the Alignment vs. Performance column for each forecaster to illustrate how representational similarity correlates with forecasting error.
3. Perturbation experiments: This analysis evaluates the effect of different parameters, such as input noise and sequence length, on both prediction accuracy and representational alignment across architectures.

**Strengths:**

I found this work to have the following strengths:
* *Clarity:* The main motivation for understanding the geometry of learned latent dynamics across different model families is well explained, with a clear limitation of prior work. The paper also highlights the limitations of current representational similarities metrics and identifies a gap in comparing forecasting systems. Later, the introduction of anchor-based relative embeddings, how they selected subset from a dataset and use the same set for computing similarity with latent representation from each model is clearly presented.
* *Originality:* The idea of using anchor-based relative embeddings to neural forecasters, is quite novel conceptually. Since, the forecasting accuracies may vary with respect to different architectures and their underlying latent state representation may differ, this paper examines the representational alignment across architectures using relative embeddings and provides architecture-specific behaviours.
* *Significance:* This work is significant in that it contributes to a better understanding of the underlying changes in latent representations in neural dynamics while forecasting trajectories. It helps clarify how different architectures behave internally and whether their representation alignment is similar or different when achieving strong forecasting accuracy. Overall, the adaptation of the anchor-based approach is quite novel and meaningful for comparing different architectures and interpreting architectural behaviors.

**Audience:**

Yes

**Audience Explanation:**

Yes.
The findings of this paper would be of interest to a meaningful portion of TMLR’s journal.

**Claims And Evidence:**

Yes

**Claims Explanation:**

The authors provides strong empirical evidence for its main claims through experiments on seven canonical dynamical systems and four neural forecaster families. However, from my perspective, the primary weaknesses of this study arise from the lack of empirical validation on real world datasets.

* *Lack of real-world datasets:* The neural forecasters in this study are primarily validated on seven canonical systems 	but does not test real-world forecasting datasets. This implies that the findings may not generalize to noisy, high-dimensional real-world time series. I strongly recommend that the authors evaluate the neural forecasters on existing real-world datasets and compare their forecasting performance on established benchmarks. This would strengthen the empirical validity of the work and demonstrate whether the representational alignment findings generalize beyond synthetic dynamical systems.
* *Anchors are dataset specific*: Although the anchor-based relative embedding idea is interesting, all the anchors are selected from a subset of dataset and used as fixed reference vectors to measure similarity with model latents. The paper does not discuss why a fixed subset of anchors is required, how results might change when anchors are selected from different datasets, or whether anchors can be learned optimally. The authors do not provide criteria or justification for their anchor selection strategy.
* *Cross-modal comparison is indirect*: Although authors mentioned that their primary goal is to compare similarities between internal representations of different models, the method compares models only indirectly through their relative similarities to a shared set of anchors, rather than through direct latent-space alignment. While this design is intentionally geometry-agnostic, the resulting similarity measures depend heavily on anchor-point relationships. Therefore, the authors should further interpret what these anchor points represent, how they influence alignment conclusions, and whether anchor selection affects the observed cross-model similarity patterns.
* *Limited discussion on results*: Although the paper reports some interesting findings such as RNNs and MLPs showing strong within-family alignment, Transformers and ESNs result in weak alignment despite high accuracy. However, the authors have not discussed why these patterns occur. In particular, because Transformer models dominate current research trends, a deeper interpretation of why their latent representations diverge from other architectures, and why they achieve higher forecasting accuracy despite low alignment, is missing. The absence of such analysis leaves an important gap in understanding architecture-specific behaviors.

For a complete and detailed account of both major and minor issues, please refer to the “Requested Changes” section.

**Requested Changes:**

Specifically, there are several points that I believe require further attention/work. I have divided these into major issues, which should be prioritized, and minor ones, which should be addressed for a strong version of current work.

**Major Comments/Questions:**
1. *The Transformer architecture is under-specified:* The authors clearly specified in the Table 1 and Table 2 that RNNs are autoregressive while Transformer uses casual attention. However, it is unclear whether the Transformer in this paper is functionally autoregressive in the forecasting sense (i.e., producing future steps sequentially) or whether it performs one-shot multi-step prediction from a single latent vector. Autoregressive Transformers typically include self-attention, feed-forward layers, and positional encodings that preserve temporal order, but the paper does not fully describe how these components are used for forecasting in this setup. This raises an important question: Which specific components of the Transformer architecture lead to high forecasting accuracy despite extremely low latent alignment? Furthermore, it is unclear whether this mismatch arises from a limitation of the relative-embedding method (e.g., anchor-based comparisons not capturing attention-based representations) or from the representational geometry of Transformers themselves, which may not form smooth latent trajectories. Clarifying these points is essential to interpret the alignment results.
2. All Figures (e.g., Fig. 1, Fig. 2, and Fig. 3) use very small font sizes for axis ticks and labels, making them hard to read. Also, none of the Figures have proper x-axis, y-axis and color legends, making it difficult to understand the results without repeatedly referring back to the text.

**Minor Comments/Typos**:
* While addressing the following points may not be critical to the paper’s core contributions, doing so would enhance the overall quality.
In Section 4.3, the authors report parameters and experimental details by referring directly to a .csv file. I recommend that all key hyperparameters and training details be described explicitly in the text rather than requiring readers to inspect external files. Since the authors already provide a GitHub repository, including a clear summary of experimental configurations within the paper would improve clarity and self-containment.
* I recommend that the authors provide a notation table summarizing the symbols used for trajectories, relative embeddings, anchor points, and other key mathematical elements. This would greatly improve readability and help readers follow the methodological components of the paper more easily.
* Although the paper evaluates models on seven canonical dynamical systems, the descriptions provided are minimal. I recommend that the authors clearly explain the differences between these systems and outline the key dynamical properties of each (e.g., periodic, quasi-periodic, chaotic, spatiotemporal chaos). This would help readers understand why these particular systems were chosen, what regimes they represent, and how they relate to the reported alignment behavior.

---

> ### Author Response · Authors · 2025-12-17
> **Architectural clarification, robustness analyses, and scope**
>
> We thank the reviewer for the detailed and constructive feedback, and for the positive assessment of the clarity, scope, and empirical coverage of the study. Below we address the main points raised and outline corresponding revisions.
>
> **Transformer specification and interpretation.**
>
> We agree that the Transformer forecaster architecture requires clearer specification. In the revised manuscript, we will explicitly clarify that our model uses a standard encoder–decoder Transformer with positional encodings and causal masking, and performs block (one-shot) multi-step prediction rather than autoregressive latent-state evolution. While causal masking enforces temporal ordering in the prediction heads, it does not induce a recurrent hidden state or a trajectory-like latent evolution.
> We will make this distinction explicit in the methods section and expand the discussion to clarify that the observed dissociation between forecasting accuracy and representational alignment in Transformers likely reflects how attention-based models summarize temporal context, rather than a limitation of the relative-embedding alignment method itself.
>
> **Figure legibility**
>
> We agree that figure readability is important. In the revised version, we will improve font sizes, axis labeling, and legend clarity in Figures 1–3. Where subplots share axes or color scales, we will ensure that labels and color bars are sufficiently clear without unnecessary visual clutter.
>
> **Experimental details and notation**
>
> To improve self-containment and readability, we will:
> add a notation table summarizing the main symbols used (trajectories, latent states, anchors, and similarity measures),
> move key hyperparameters and training details from external files into the appendix,
> and include brief descriptions of each dynamical system (periodic, quasi-periodic, chaotic, spatiotemporal) to better motivate their selection and contextualize the reported alignment behavior.
>
> **Additional clarifications**
>
> **Anchor-based relative embeddings and anchor selection**
>
> We use a standard anchor-based relative embedding framework to obtain geometry-agnostic representations that enable cross-architecture comparison by removing rotational and scaling ambiguities. We do not claim that this approach is optimal; rather, it provides a pragmatic and well-established relational reference frame. We note that several follow-up and complementary methods have been proposed: for example by improving robustness through spectral formulations (Fumero et al., 2025), estimating direct transformations between latent spaces (Maiorca et al., 2023), or constructing shared product latent spaces induced by multiple invariant similarity functions (Cannistraci et al., 2024).
>
> To address concerns about anchor dependence, we include an anchor ablation study in the Appendix Fig. 9 showing that alignment estimates stabilize as the number of anchors increases. In the revision, we will further clarify the role of anchors, discuss design trade-offs (e.g., stability vs. computation), and explicitly acknowledge limitations related to anchor choice.
>
> **Similarity metrics**
>
> We agree that robustness to the choice of similarity metric is important. In the revised manuscript, we include explicit comparisons between our primary metric (cosine similarity on anchor-based relative embeddings) and standard representational similarity measures, including RSA, Procrustes-based alignment, and CKA, evaluated on the same models and systems.
> We find that the main qualitative conclusions—most notably the family-level alignment structure and the dissociation between forecasting accuracy and representational alignment in Transformers and ESNs—are consistent across these measures. We will add a dedicated paragraph and corresponding Appendix figures to make this comparison explicit.
>
> **Scope and interpretation**
>
> We acknowledge that our evaluation focuses on low-dimensional canonical dynamical systems chosen for control and interpretability. We will strengthen the discussion of this scope and its limitations in the revised manuscript, and clarify the regimes in which our conclusions are intended to apply. Extending the analysis to higher-dimensional or real-world datasets is an important direction for future work, and where feasible we will explore such extensions during the revision.
>
> We thank the reviewer again for the constructive feedback, which has helped us improve the clarity, positioning, and completeness of the manuscript.

---

> > ### Comment · Reviewer_pBzn · 2026-01-14
> >
> > I thank the authors for addressing my major and minor comments. However, questions regarding validation on real-world datasets and deeper discussion of the experimental results, as raised in the weaknesses section, remain insufficiently addressed. Additionally, Figure 2 and 3 still lack axis labels, and the captions does not clearly specify what the axes represent, which limits interpretability.
> >
> > Incorporating these aspects would significantly strengthen the paper. Overall, the current version is in good shape, but addressing the above points would result in a stronger and more convincing submission.

---

> > > ### Author Response · Authors · 2026-01-14
> > > **Clarifications and Revisions**
> > >
> > > Thank you for the follow-up and for noting that the manuscript is in good shape overall.
> > >
> > > Regarding validation on real-world data, we would like to clarify that we have now added a high-dimensional real-world experiment using intracranial EEG (iEEG) data in Appendix F (Fig. 15). These results are explicitly labeled as preliminary and are included as an external validation, showing that the main family-level alignment patterns observed in canonical dynamical systems persist in a real-world forecasting setting.
> > >
> > > Regarding figure readability, we agree that clarity can be further improved. In the final revision, we will add explicit axis labels to all subplots in Figures 2 and 3 and expand the figure captions to clearly specify what each axis represents, even when axes are shared across panels.
> > >
> > > We thank the reviewer again for the constructive feedback, which has helped improve the clarity and completeness of the presentation.

---

### Review · Reviewer_t9fv · 2025-12-08

**Summary Of Contributions:**

**Summary**

This paper studies representational alignment in neural forecasters for dynamical systems. The authors use a common encoder, propagator , decoder forecasting framework, and instantiate it with several model families such as multilayer perceptrons, recurrent networks, Transformers and echo state networks. They adopt an anchor-based relative embedding method to transform latent states and ground truth trajectories into geometry aware representations, and define similarity metrics between models on this basis. Experiments on seven canonical dynamical systems compare model families, evaluate how alignment relates to forecasting error under different noise levels and context lengths, and analyze how well encoders and decoders can be stitched across models. The paper reports consistent family level structure in the learned latent spaces, shows that high forecasting accuracy can appear together with low alignment especially for Transformers and echo state networks, and argues that relative geometry offers a stable tool to compare neural forecasters beyond standard loss based evaluation.

**Main Strengths**

1. The paper provides a systematic empirical study of representational alignment in neural forecasters across multiple model families and dynamical systems. The experimental design is thorough with a shared framework, comparable training setups, and a rich set of visualizations. This gives the community a consolidated view of how different architectures organize their latent spaces in this setting.

2. The use of anchor-based relative embeddings offers a clear and unified lens to compare representations from heterogeneous models. By mapping both latent states and ground truth trajectories into a common relative space, the paper avoids trivial invariances and allows model to model comparisons.

3. The work reveals several interesting empirical phenomena that can inspire further research. Examples include family level structure in the learned geometries and the observation that high forecasting accuracy can coexist with low alignment for some architectures.
These findings raise concrete questions about inductive biases and optimization that go beyond standard loss based evaluation.

**Main Weaknesses**

1. The forecasting architectures are standard and the relative embedding framework closely follows prior work with only light adaptation to dynamical systems.

2. The evaluation relies almost entirely on cosine similarity in the relative embedding space. Other similarity measures and a more systematic robustness analysis are missing, so the choice of metric is not fully justified.

3. The analysis of differences between model families is mostly descriptive. The paper reports patterns for MLPs, RNNs, Transformers, and ESNs but offers little mechanistic explanation or theory to support these observations.

4. Latent states are treated as black box vectors, with no probing experiments or strong links to physical quantities or dynamical properties, which limits the depth of the conclusions about representation geometry.

**Audience:**

Yes

**Audience Explanation:**

I think the work speaks to readers who care about neural forecasting for dynamical systems. The study offers a systematic comparison of common forecaster architectures and highlights family level structure in their latent spaces. Researchers interested in neural ODEs, Koopman operators, and model interpretability may find the relative geometry analysis useful as a diagnostic tool.

**Broader Impact Concerns:**

1. The study focuses on representation analysis for neural forecasters on synthetic or standard dynamical systems.
It would still be useful to add a short Broader Impact statement that clarifies this low risk profile.

2. In those settings misinterpretation of model reliability or alignment metrics could influence high impact decisions. The authors could briefly discuss the need for domain specific validation before using these tools in real-world decision-making.

**Claims And Evidence:**

Yes

**Claims Explanation:**

The main claims are supported by systematic experiments on several canonical dynamical systems with multiple neural forecaster families. The methodology for relative embeddings and alignment is described in sufficient detail and is implemented consistently across models. Some broader interpretive statements, such as the generality of relative geometry as a tool for representation science, are more qualitative, but they remain plausible.

**Requested Changes:**

1. The paper relies almost entirely on cosine similarity in the relative embedding space. I recommend adding comparisons with other similarity measures such as CKA, Procrustes based metrics, or RSA on the same setups. Please also study robustness to the choice and number of anchors in a more systematic way, for example through ablations and statistical summaries.

2. The current analysis mainly describes that different architectures show different alignment and accuracy patterns. It would be important to provide a more mechanistic explanation that links these patterns to properties of MLPs, RNNs, Transformers, and ESNs.

3. At present the latent states are treated as black box vectors. I suggest adding probing experiments that connect representations to physical quantities, dynamical invariants, or local stability indicators. Such analyses would provide solid evidence about what information the latent geometry actually encodes and would greatly strengthen the main claims.

4. The paper argues that relative geometry can help evaluate and compare forecasters, yet it does not show concrete decision scenarios.
I recommend adding at least one experiment where alignment metrics are used for early stopping, model selection, or failure detection.
Demonstrating that these metrics change a practical choice would make the contribution more compelling for practitioners.

5. Several statements present the framework as a simple and general tool for representation science, which may feel stronger than what the experiments support. Please better situate the work relative to prior alignment methods and clarify the specific regimes where the proposed approach is most appropriate. A more explicit discussion of limitations and of settings where the method may fail would improve the balance and credibility of the paper.

---

> ### Author Response · Authors · 2025-12-17
> **Robustness analyses, mechanistic interpretation, and probing**
>
> We thank the reviewer for the thoughtful and constructive assessment, and for recognizing the systematic nature of the empirical study and the value of relative geometry as a comparative diagnostic across neural forecasters. We address the requested changes below and outline corresponding revisions.
>
> **Comparison with other similarity measures and anchor robustness**
>
> We agree that robustness to the choice of similarity metric is important. In the revised manuscript, we include explicit comparisons between our primary metric (cosine similarity on anchor-based relative embeddings) and standard representational similarity measures, including RSA, Procrustes-based alignment, and CKA, evaluated on the same models and systems. We find that the main qualitative conclusions—most notably the family-level alignment structure and the dissociation between forecasting accuracy and representational alignment for Transformers and ESNs—are consistent across these measures.
>
> Regarding anchor dependence, we already include an anchor ablation study in Appendix Fig. 9 showing that alignment estimates stabilize beyond a moderate number of anchors. In the revision, we will make this analysis more prominent to clarify stability–variance trade-offs.
>
> **Mechanistic explanation of family-level alignment patterns**
>
> We agree that a more explicit interpretation of the observed family-level patterns is valuable. In the revised manuscript, we will expand the discussion to connect these patterns to architectural inductive biases within the shared encoder–propagator–decoder framework.
>
> RNN-based forecasters maintain a recursively updated hidden state, which induces temporally coherent latent trajectories and leads to consistently high alignment across seeds and variants. MLP forecasters compress each input window into a single global latent via a fixed feedforward mapping, resulting in simpler but relatively stable within-family geometry. In contrast, Transformer encoders construct token-wise contextual representations in parallel via self-attention, without architectural pressure to form smooth latent trajectories, enabling high forecasting accuracy with weaker representational alignment. ESNs rely on fixed random reservoirs, and since only the readout is trained, reservoir trajectories primarily reflect internal reservoir dynamics rather than the target system, explaining their systematically lower alignment.
>
> These explanations remain qualitative, but help situate the empirical findings in terms of architectural design choices.
>
> **Probing and links to physical or dynamical quantities**
>
> In response to the reviewer’s suggestion, we will add a lightweight probing analysis that decodes the observable system state x(t) from latent representations z(t) using linear ridge regression, evaluated on a representative system (Lorenz-63). For absolute latent representations, decoding performance is near-perfect across model families, while for relative (anchor-based) embeddings decoding remains strong but exhibits a modest, architecture-dependent reduction.
>
> We interpret this probe as a sanity check demonstrating that state information is present in the learned latent representations. At the same time, we emphasize that anchor-based relative embeddings are not designed to preserve all linearly decodable structure, nor do we make general claims about linear probe performance beyond the specific experimental setting considered here.
>
> We will include these probing results in the appendix and discuss them as a boundary on interpretation rather than as evidence of system identification or dynamical recovery. Probing analyses targeting invariants, stability properties, or governing equations would require a substantially different, more physics-oriented methodological focus and are therefore left for future work.

---

> > ### Author Response · Authors · 2025-12-19
> > **Missing Answer to the Last Two Points**
> >
> > We realised that we did not post our full response. We apologise and add our response to the last two points below.
> >
> > **Use of alignment as a diagnostic tool**
> > We appreciate the reviewer’s suggestion to explore alignment-guided early stopping or model selection. However, we intentionally do not make such claims in this work. Our goal is to characterise and compare representational geometry induced by forecasting objectives, not to propose alignment as a training or decision-making criterion.
> > That said, Fig. 2 already illustrates that alignment evolves during training in ways that are not captured by prediction error alone and differ systematically across architectures. We therefore frame alignment as a descriptive signal that reveals representational properties and inductive biases during training, rather than as a prescriptive tool. We will make this distinction explicit in the revised manuscript and clarify that practical use of alignment for training control remains an open direction for future work.
> >
> >
> > **Positioning, limitations, and broader impact**
> > We will better situate the contribution relative to prior alignment methods and clarify the regimes where anchor-based relative geometry is appropriate—namely, comparative analysis across heterogeneous models and seeds under controlled settings. We will also expand the discussion of limitations, including dependence on anchor choice, focus on synthetic systems, and abstraction away from physical state recovery. Finally, we will add a brief Broader Impact statement clarifying the low-risk nature of the work and emphasizing that domain-specific validation would be required before using alignment metrics in real-world decision-making.

---

### Review · Reviewer_6aaX · 2026-01-09

**Summary Of Contributions:**

The paper investigates representational alignment in neural networks that forecast dynamics.
This is achieved using anchor-based relative embeddings.
The authors systematically compare latent space geometries across model families (MLPs, RNNs, Transformers, ESNs) on seven canonical dynamical systems spanning periodic, quasi-periodic, and chaotic regimes.
The key finding is that architectural families exhibit reproducible internal alignment patterns, and importantly, that high forecasting accuracy can coexist with low representational alignment, particularly for Transformers and ESNs.
The work adapts the relative embedding framework of Moschella et al. (2023) to the setting of dynamical systems forecasting and provides extensive empirical analysis.

**Strengths**
- Addresses an important and underexplored question: when do forecasting nets learn structurally similar representations of dynamics?
- Systematic evaluation across diverse dynamical regimes and architectures with strong reproducibility
- The finding that accuracy and alignment can diverge has practical implications for assessing the robustness of models.
- Clear experimental framework with thorough analyses
- Well-executed, reproducible empirical study

**Weaknesses**
- Limited exploration of when alignment matters for downstream tasks (generalization, robustness)
- Anchor selection is not deeply investigated
- The evaluation relies solely on short-horizon MSE, missing richer dynamical metrics (attractor geometry, Lyapunov exponents, long-term statistics) that could reveal deeper connections between alignment and faithful dynamical reconstruction.
- No theoretical grounding

**Overall:**
A valuable empirical contribution that opens an important research direction.
The framework may not tell us whether models learn the underlying true dynamics, but it provides a principled tool for assessing whether models learn consistent dynamics, which is crucial for robustness, model comparison, and scientific understanding.

**Additional Comments:**

**Optional Discussion Points**

The following are optional changes/points that the authors are encouraged to add in their discussion.
I raise them in the spirit of scientific dialogue, as I find the paper particularly interesting.

**On Loss Functions and Dynamical Systems Evaluation Metrics**:
The paper focuses on MSE as the forecasting loss, which is standard and reasonable. However, the dynamical systems community has developed a rich set of evaluation metrics that go beyond short-horizon MSE and capture different aspects of learned dynamics:
- **Attractor geometry**: Do the predicted trajectories lie on/near the true attractor? (e.g., correlation dimension, fractal dimension)
- **Spectral properties**: Do power spectra or Lyapunov exponents match?
- **Long-term statistics**: Climate/invariant measure metrics (do long rollouts reproduce the correct statistical distribution over state space?)
- **Topological properties**: Are qualitative features (fixed points, limit cycles, basins of attraction) preserved?

It would be interesting to explore whether representational alignment correlates differently with these alternative evaluation metrics.
For example, a model might achieve a low MSE but poor long-term climate performance.
Does alignment predict this failure mode?
Conversely, models with similar alignment might share similar long-term statistical behavior even if short-term errors differ.
This could connect the representational analysis to the broader goals of faithful dynamical reconstruction, though I recognize this would be a substantial extension.

**On Architecture vs. Training Algorithm Effects**:
The current study attributes the observed alignment patterns primarily to architectural inductive biases (MLPs vs. RNNs vs. Transformers vs. ESNs), and all models are trained with the same optimization procedure (Adam, MSE loss, early stopping).
This raises a natural question: to what extent are the alignment patterns a property of the architecture alone, or a joint effect of the architecture and the training algorithm?

For instance:
- Would different optimizers (SGD, second-order methods) or learning rate schedules produce different alignment patterns for the same architecture? (probably not)
- Training strategies designed for long-term forecasting (such as curriculum learning over prediction horizons, scheduled sampling, or backpropagation through longer rollouts) might encourage latent representations that better capture long-term dynamics. Would such training schemes lead to higher alignment with the true system? (no intuition here)
- Regularization techniques (e.g., noise injection, spectral normalization) might implicitly shape the geometry of learned representations.

It is plausible that alignment is primarily architecture-driven, but it would be valuable to disentangle architectural from algorithmic contributions. Even a preliminary experiment comparing, say, standard training versus a long-horizon-aware training scheme on the same architecture could shed light on this. This might also connect to practical recommendations: if alignment can be improved through training modifications, it becomes an actionable target rather than a passive diagnostic.

**On Anchor Selection and Optimal Experimental Design**:
The current approach selects anchors randomly from the dataset, and the ablation shows stability beyond a moderate number. However, for dynamical systems, one wonders whether there is a principled way to select anchors that would be more informative or efficient. For instance:
- Should anchors be chosen to sparsely but optimally cover the latent attractor or state-space region, e.g., clustering on the attractor, or selecting points near dynamically significant regions (saddle points, attractor boundaries, transient vs. steady-state regions)?
- Is there a notion of "anchor efficiency", i.e., achieving stable alignment estimates with fewer, more strategically chosen anchors?
This is likely beyond the scope of the current work, but could be a fruitful direction for follow-up studies, particularly for high-dimensional systems where anchor count affects computational cost.

**On the relation with the Platonic representation hypothesis [1]**
The Platonic Representation Hypothesis argues that as models scale and become more capable, their representations converge toward a shared statistical model of reality.
The hypothesis was formulated based on large-scale foundation models in vision and NLP trained on internet-scale data for general-purpose tasks, a setting quite different from the task-specific neural forecasters studied here.
Nonetheless, the connection is interesting: while the Platonic hypothesis predicts convergence with scale, this paper shows that in forecasting, architectures (of much smaller scale) can achieve similar accuracy while their representations diverge, suggesting that task performance alone does not guarantee representational alignment.

[1] Minyoung Huh et. al. "The Platonic Representation Hypothesis"

**Audience:**

Yes

**Audience Explanation:**

Yes. The paper may not introduce any new representation learning method, but the findings are genuinely interesting for anyone working on forecasting, whether in dynamical systems, weather, fluid flows, or similar fields.
The observation that models can forecast accurately while learning very different internal representations raises important questions about what these models actually learn.
The analysis is a solid starting point for understanding how different architectures organize their latent space and could pave the way for further work that connects representational structure to generalization, robustness, etc.

**Important and Timely Question**:
The paper addresses a fundamental question that is often overlooked in the neural forecasting literature: when do two models learn structurally similar representations of dynamics?
This is distinct from and complementary to the standard question of forecasting accuracy.
The community has focused heavily on prediction error metrics, but understanding *how* models internally represent temporal structure is crucial for interpretability, robustness, and trustworthiness.

**Practical Relevance Beyond Academia**: The finding that models can achieve similar forecasting accuracy while learning structurally different latent representations has significant implications for industrial applications. In a wide range of applications where neural forecasters are applied (from weather and climate prediction, energy grid management, and financial markets to fluid dynamics, structural health monitoring, and digital twins, just to name some), practitioners need to understand not just whether a model works, but also how it works and whether its learned representations are robust. The proposed framework provides a tool for:
   - Detecting when model updates or retraining lead to structurally different solutions
   - Evaluating robustness: models with similar alignment patterns may generalize more consistently
   - Model selection beyond accuracy: preferring models whose representations align with physical intuition or other trusted models

**Broader Impact Concerns:**

No significant concerns.

**Claims And Evidence:**

Yes

**Claims Explanation:**

Yes. The paper shows that model families exhibit reproducible internal alignment patterns (MLPs align with MLPs, RNNs with RNNs) while also demonstrating that high forecasting accuracy can coexist with low alignment to other models or to the ground-truth trajectory structure, particularly for Transformers and ESNs.
These findings are consistent across seven dynamical systems, multiple random seeds, and various metrics.
The experiments are thorough, and the conclusions are appropriately cautious.

**Requested Changes:**

The authors have performed a thorough revision addressing the main concerns raised by reviewers, and I find the paper adequate for publication.
I offer below some optional discussion points for the authors that, in my opinion, would strengthen the work.

---

### Decision · Action_Editor_66Lj · 2026-01-14

**Recommendation:** Accept with minor revision

**Additional Comments:**

Since not all announced revisions where directly added to the revised manuscript in the short time of the discussion phase, please go again through the planned additions and optional points, and consider what further improvements could be included in the camera-ready version.

In particular, I ask the authors to consider the comments of Reviewer 6aaX who joined the reviewer pool at a late stage. You may still write a response comment to 6aaX, (though this is not strictly required).

Please also implement the small improvements to the figures as agreed with Reviewer pBzn.

**Audience:**

Yes

**Audience Explanation:**

The submission addresses a relevant and unsolved question about forecasting, in how far high accuracy implies that models have learned similar latent dynamics. All reviewers agreed on the overall interest of the findings.

**Claims And Evidence:**

Yes

**Claims Explanation:**

The claims made in the submission where found to be overall well supported by experiments, including additional analysis of the method to assess robustness. The reviewers agreed on the technical soundness of the paper, although it was pointed out that a mechanistic explanation of the findings or additional analysis of long-horizon statistics would strengthen the results.

All reviewers felt that their main concerns were addressed in the revision phase and recommended acceptance, with some recommending optional additions.

---

> ### Author Response · Authors · 2026-01-21
> **Final Revisions and Implementation of Author Commitments**
>
> Dear Action Editor,
>
> Thank you very much for your careful handling of the review process and for your guidance throughout.
>
> We are currently working on the final modifications to the manuscript and will ensure that all changes and additions we explicitly committed to in our responses are fully and consistently implemented in the camera-ready version. In particular, we will incorporate the agreed-upon figure improvements and carefully consider the optional discussion points raised by Reviewer 6aaX.
>
> We appreciate your time and support in overseeing the review and revision process.
>
> Kind regards,
> The Authors